# Toward Understanding Generative Data Augmentation

**Chenyu Zheng**[1,2]**, Guoqiang Wu**[3]**, Chongxuan Li**[1,2*]
[1] Gaoling School of Artificial Intelligence, Renmin University of China, Beijing, China
[2] Beijing Key Laboratory of Big Data Management and Analysis Methods, Beijing, China
[3] School of Software, Shandong University, Shandong, China
{chenyu.zheng666, guoqiangwu90}@gmail.com; chongxuanli@ruc.edu.cn

## Abstract

Generative data augmentation, which scales datasets by obtaining fake labeled examples from a trained conditional generative model, boosts classification performance in various learning tasks including (semi-)supervised learning, few-shot learning, and adversarially robust learning. However, little work has theoretically investigated the effect of generative data augmentation. To fill this gap, we establish a general stability bound in this not independently and identically distributed (non-i.i.d.) setting, where the learned distribution is dependent on the original train set and generally not the same as the true distribution. Our theoretical result includes the divergence between the learned distribution and the true distribution. It shows that *generative data augmentation can enjoy a faster learning rate when the order of divergence term is* $o(\max\left(\log(m)\beta_m, 1/\sqrt{m}\right))$, where $m$ is the train set size and $\beta_m$ is the corresponding stability constant. We further specify the learning setup to the Gaussian mixture model and generative adversarial nets. We prove that *in both cases, though generative data augmentation does not enjoy a faster learning rate, it can improve the learning guarantees at a constant level when the train set is small, which is significant when the awful overfitting occurs.* Simulation results on the Gaussian mixture model and empirical results on generative adversarial nets support our theoretical conclusions. Our code is available at *https://github.com/ML-GSAI/Understanding-GDA*.

## 1 Introduction

Deep generative models [1; 2; 3; 4] have achieved great success in many fields, including computer vision [5; 6], natural language processing [7; 8; 9], and cross-modal learning [10; 11; 12] in the recent years. A promising usage built upon them is generative data augmentation (GDA) [13; 14; 15], which scales the train set by producing synthetic examples with labels based on advanced conditional generative models. Empirically, it has been observed that GDA can improve classification performance in lots of settings, including supervised learning [16; 17], semi-supervised learning [18; 19; 20], few-shot learning [21], zero-shot learning [22], adversarial robust learning [23; 24], etc.

Although promising algorithms and applications of GDA emerge in different learning setups, our experiments in Section 4 show that GDA does not always work, such as in the case with a rich train set or standard augmentation methods (e.g., flip). Besides, the number of augmented data has a significant impact on the performance while is often tuned manually. These phenomena motivate us to study the effect of GDA. Unfortunately, little work has investigated this technique from a theoretical perspective. Therefore, in this paper, we take a first step towards understanding it. Specially, we consider the supervised classification setting, and try to answer the following questions rigorously:

- *Can we establish learning guarantees for GDA and explain when it works precisely?*

---

*Correspondence to Chongxuan Li.

- *Can we obtain theoretical insights on hyperparameters like the number of augmented data?*

Our first main contribution is to propose a general algorithmic stability bound for GDA in Section 3.1. The main technical challenge is that GDA breaks the primary i.i.d. assumption of the classical results [25; 26] because the distribution learned by the generative model is dependent on the sampled train set and generally not the same as the true distribution. Besides, it is unclear whether the existing general non-i.i.d. stability bounds [27; 28; 29] are suitable to derive meaningful guarantees for GDA. Informally, our result (Theorem 3.1) can be presented as follows:

$$|\textit{Gen-error}| \lesssim \text{distributions' divergence} + \text{generalization error w.r.t. mixed distribution},$$

where *Gen-error* means the generalization error of GDA, and $a \lesssim b$ means $a = O(b)$. The distributions' divergence term on the right hand is caused by the divergence between the distribution learned by the generative model and the true distribution. In addition, the remaining generalization error w.r.t. mixed distribution vanishes as we increase the augmentation size. Comparing this bound to the classical result without GDA (Theorem 2.1), we can obtain an exact condition for GDA to be effective: *GDA can enjoy a faster learning rate when the order of divergence term is* $o(\max\left(\log(m)\beta_m, 1/\sqrt{m}\right))$, where $m$ is the train set size and $\beta_m$ is the corresponding uniform stability constant. This means the performance of the chosen generative model matters a lot.

Our second main contribution is to particularize the general results to the binary Gaussian mixture model (bGMM) and generative adversarial nets (GANs) [2] in Section 3.2 and Section 3.3, respectively. Our theoretical results (Theorems 3.2 and 3.3) show that, in both cases, the order of the divergence term in the obtained upper bound is $\Omega(\max\left(\log(m)\beta_m, 1/\sqrt{m}\right))$. Therefore, when the train set size is large enough, it is hopeless to use GDA to boost the classification performance by a large margin. Worse still, GDA may damage the generalization of the learning algorithm. However, *when the train set size is small and awful overfitting happens, GDA can improve the learning guarantee at a constant level, which is significant in this situation*. These theoretical implications show the promise of GDA in real-world problems with limited data.

Finally, experiments presented in Section 4 validate our theoretical findings. In particular, in the bGMM setting, experimental results show that our generalization bound (Theorem 3.2) predicts the order and trend of true generalization error well. Besides, in our empirical study on the real image dataset, we find that GANs can not boost the test performance obviously and even damage the generalization when standard data augmentation methods are used to approximate the case with a large train set. In contrast, GANs improve the performance by a large margin when the train set size is small and terrible overfitting occurs. All these experimental results support our theoretical implications in Section 3. Furthermore, we also conduct experiments with the state-of-the-art diffusion model [30]. Empirical results show the promise of the diffusion model in GDA and suggest it could have a faster learning rate than GAN.

## 2 Preliminaries

### 2.1 Notations

Let $\mathcal{X} \subseteq \mathbb{R}^d$ be the input space and $\mathcal{Y} \subseteq \mathbb{R}$ be the label space. We denote by $\mathcal{D}$ the population distribution over $\mathcal{Z} = \mathcal{X} \times \mathcal{Y}$. The $L_p$ norm of a random variable $X$ is denoted as $\|X\|_p = (\mathbb{E}|X|^p)^{\frac{1}{p}}$. Given a set $S = \{\mathbf{z}_1, \mathbf{z}_2, \ldots, \mathbf{z}_m\}$, we define $S^{\backslash i}$ as the set after removing the $i$-th data point in the set $S$, and $S^i$ as the set after replacing the $i$-th data point with $\mathbf{z}'_i$ in the set $S$. Let $[m] = \{1, 2, \ldots, m\}$, then for every set $V \subseteq [n]$, we define $S_V = \{\mathbf{z}_i : i \in V\}$. In addition, for some function $f = f(S)$, we denote its conditional $L_p$ norm with respect to $S_V$ by $\|f\|_p (S_V) = (\mathbb{E}[\|f\|^p \mid S_V])^{\frac{1}{p}}$. Besides, we denote the total variation distance by $\mathcal{D}_{\text{TV}}$ and KL divergence by $\mathcal{D}_{\text{KL}}$, respectively.

We let $(\mathcal{Y})^{\mathcal{X}}$ be the set of all measurable functions from $\mathcal{X}$ to $\mathcal{Y}$, $\mathcal{A}$ be a learning algorithm and $\mathcal{A}(S) \in (\mathcal{Y})^{\mathcal{X}}$ be the hypothesis learned on the dataset $S$. Given a learned hypothesis $\mathcal{A}(S)$ and a loss function $\ell : (\mathcal{Y})^{\mathcal{X}} \times \mathcal{Z} \to \mathbb{R}_+$, the true error $\mathcal{R}_{\mathcal{D}}(\mathcal{A}(S))$ with respect to the data distribution $\mathcal{D}$ is defined as $\mathbb{E}_{\mathbf{z} \sim \mathcal{D}}[\ell(\mathcal{A}(S), \mathbf{z})]$. In addition, the corresponding empirical error $\widehat{\mathcal{R}}_S(\mathcal{A}(S))$ is defined as $\frac{1}{m} \sum_{i=1}^m \ell(\mathcal{A}(S), \mathbf{z}_i)$.

## 2.2 Generative data augmentation

In this part, we describe the process of GDA in a mathematical way. Given a training set $S$ with $m_S$ i.i.d. examples from $\mathcal{D}$, we can train a conditional generative model $G$, and denote the model distribution by $\mathcal{D}_G(S)$. We note that the randomness from training the generative model is ignored in this paper. In addition, we define the expectation of the model distribution with regard to $S$ as $\mathcal{D}_G = \mathbb{E}_S[\mathcal{D}_G(S)]$. Based on the trained generative model, we can then obtain a new dataset $S_G$ with $m_G$ i.i.d. samples from $\mathcal{D}_G(S)$, where $m_G$ is a hyperparameter. Typically, we consider the case that $m_G = \Omega(m_S)$ if GDA is utilized. We denote the total number of the data points in augmented set $\widetilde{S} = S \cup S_G$ by $m_T$. Besides, we define the mixed distribution after augmentation as $\widetilde{\mathcal{D}}(S) = \frac{m_S}{m_T}\mathcal{D} + \frac{m_G}{m_T}\mathcal{D}_G(S)$. As a result, a hypothesis $\mathcal{A}(\widetilde{S})$ can be learned on the augmented dataset $\widetilde{S}$. To understand the effect of GDA, we are interested in the generalization error $|\mathcal{R}_{\mathcal{D}}(\mathcal{A}(\widetilde{S})) - \widehat{\mathcal{R}}_{\widetilde{S}}(\mathcal{A}(\widetilde{S}))|$ with regard to the learned hypothesis $\mathcal{A}(\widetilde{S})$. For convenience, we denote it by *Gen-error* in the remaining paper. Technically, we establish bounds for *Gen-error* using the algorithmic stability introduced in the next subsection. As far as we know, this is the first work to investigate the learning guarantees for GDA.

## 2.3 Generalization via algorithmic stability

Algorithmic stability analysis is a important tool to provide guarantees for the generalization of machine learning models. A key advantage of stability analysis is that it exploits particular properties of the algorithm and provides algorithm-dependent bounds. Different notations of stability have been proposed and used to establish high probability bounds for generalization error [25; 31; 32; 33]. Among them, uniform stability is the most widely used and has been utilized to analyze the generalization of many learning algorithms, including regularized empirical risk minimization (ERM) algorithms [25] and stochastic gradient descent (SGD) [32; 34; 35]. The uniform stability is defined as the following.

**Definition 2.1** (Uniform stability). *Algorithm $\mathcal{A}$ is uniformly $\beta_m$-stable with respect to the loss function $\ell$ if the following holds*

$$\forall S \in \mathcal{Z}^m, \forall \mathbf{z} \in \mathcal{Z}, \forall i \in [m], \sup_{\mathbf{z}} \left| \ell(\mathcal{A}(S), \mathbf{z}) - \ell(\mathcal{A}(S^i), \mathbf{z}) \right| \leq \beta_m.$$

Given a $\beta_m$-stable learning algorithm, the milestone work [25] provides a high probability generalization bound that converges when $\beta_m = o(1/\sqrt{m})$. This condition may fail to hold in some modern machine learning settings [36], which leads to meaningless guarantees. In recent years, some works [37; 38; 26] improved the classical bound by establishing novel and tighter concentration inequalities. Especially, [26] proposed a moment bound and obtained a nearly optimal generalization guarantee, which only requires $\beta_m = o(1/\log m)$ to converge. It is listed in the next theorem.

**Theorem 2.1** (Corollary 8, [26]). *Assume that $\mathcal{A}$ is a $\beta_m$-stable learning algorithm and the loss function $\ell$ is bounded by $M$. Given a training set $S$ with $m$ i.i.d. examples sampled from the distribution $\mathcal{D}$, then for any $\delta \in (0, 1)$, with probability at least $1 - \delta$, it holds that*

$$\left| \mathcal{R}_{\mathcal{D}}(\mathcal{A}(S)) - \widehat{\mathcal{R}}_S(\mathcal{A}(S)) \right| \lesssim \log(m)\beta_m \log\left(\frac{1}{\delta}\right) + M\sqrt{\frac{1}{m}\log\left(\frac{1}{\delta}\right)}. \tag{1}$$

We note that all generalization bounds mentioned above require a primary condition: data points are drawn i.i.d. according to the population distribution $\mathcal{D}$. However, it no longer holds in the setting of GDA. On the one hand, the distribution $\mathcal{D}_G(S)$ learned by the generative model is generally not the same as the true distribution $\mathcal{D}$. On the other hand, the learned $\mathcal{D}_G(S)$ is heavily dependent on the sampled dataset $S$. This property brings obstacles to the derivation of the generalization bound for GDA. Furthermore, though there exists some non-i.i.d. stability bounds [27; 28; 29], it is still unclear whether these techniques are suitable in the GDA setting.

## 3 Main theoretical results

In this section, we present our main theoretical results. In Section 3.1, we establish a general generalization bound (Theorem 3.1) for GDA. Built upon the general learning guarantee, we then

particularize the learning setup to the bGMM introduced in Section 3.2.1 and derive a specified generalization bound (Theorem 3.2). Finally, we discuss our theoretical implications on GANs in real-world problems (Theorem 3.3). Notably, to the best of our knowledge, this is the first work to investigate the generalization guarantee of GDA.

## 3.1 General generalization bound

To understand GDA, we are interested in studying the generalization error of the hypothesis $\mathcal{A}(\widetilde{S})$ learned on the dataset $\widetilde{S}$ after augmentation. Formally, we need to bound $|\mathcal{R}_{\mathcal{D}}(\mathcal{A}(\widetilde{S})) - \widehat{\mathcal{R}}_{\widetilde{S}}(\mathcal{A}(\widetilde{S}))|$, which has been defined as *Gen-error* in Section 2.2. Recall that $\widetilde{\mathcal{D}}(S)$ has been defined as the mixed distribution after augmentation, to derive such a bound, we first decomposed *Gen-error* as

$$|\textit{Gen-error}| \leq \underbrace{\left| \mathcal{R}_{\mathcal{D}}(\mathcal{A}(\widetilde{S})) - \mathcal{R}_{\widetilde{\mathcal{D}}(S)}(\mathcal{A}(\widetilde{S})) \right|}_{\text{Distributions' divergence}} + \underbrace{\left| \mathcal{R}_{\widetilde{\mathcal{D}}(S)}(\mathcal{A}(\widetilde{S})) - \widehat{\mathcal{R}}_{\widetilde{S}}(\mathcal{A}(\widetilde{S})) \right|}_{\text{Generaliztion error w.r.t. mixed distribution}} .$$

The first term on the right hand can be bounded by the divergence (e.g., $\mathcal{D}_{\mathrm{TV}}, \mathcal{D}_{\mathrm{KL}}$) between the mixed distribution $\widetilde{\mathcal{D}}(S)$ and the true distribution $\mathcal{D}$. It is heavily dependent on the ability of the chosen generative model. For the second term, we note that classical stability bounds (e.g. Theorem 2.1) can not be used directly, because points in $\widetilde{S}$ are drawn non-i.i.d.. We mainly use a core property of $\widetilde{S}$, that is, $S$ satisfies the i.i.d. assumption, and $S_G$ satisfies the conditional i.i.d. assumption when $S$ is fixed. Inspired by this property, we furthermore decompose this term and utilize sharp moment inequalities [39; 26] to obtain an upper bound. Finally, we conclude with the following result.

**Theorem 3.1** (Generalization bound for GDA, proof in Appendix B.1). *Assume that $\mathcal{A}$ is a $\beta_m$-stable learning algorithm and the loss function $\ell$ is bounded by $M$. Given an set $\widetilde{S}$ augmented as described in Section 2.2, then for any $\delta \in (0,1)$, with probability at least $1 - \delta$, it holds that*

$$|\textit{Gen-error}| \lesssim \underbrace{\frac{m_G}{m_T} M \mathcal{D}_{\mathrm{TV}} \left( \mathcal{D}, \mathcal{D}_G(S) \right)}_{\text{Distributions' divergence}} + \frac{M(\sqrt{m_S} + \sqrt{m_G}) + m_S \sqrt{m_G} \beta_{m_T}}{m_T} \sqrt{\log\left(\frac{1}{\delta}\right)}$$

$$+ \frac{\beta_{m_T}\left(m_S \log m_S + m_G \log m_G\right) + m_S \log m_S M \mathcal{T}(m_S, m_G)}{m_T} \log\left(\frac{1}{\delta}\right),$$

*where $\mathcal{T}(m_S, m_G) = \sup_i \mathcal{D}_{\mathrm{TV}} \left( \mathcal{D}_G^{m_G}(S), \mathcal{D}_G^{m_G}(S^i) \right)$.*

*Remark.* **Tightness of the proposed upper bound.** Let $m_G = 0$, we observe that Theorem 3.1 degenerates to Theorem 2.1. Therefore, our stability bound includes the i.i.d. setting as a special case and benefits from the same nearly optimal guarantee shown by [26]. Further analysis of the tightness of our guarantee when $m_G > 0$ is left to future work.

*Remark.* **Comparison with the existing non-i.i.d. stability bounds.** Detailed introduction for non-i.i.d. stability bounds is placed in Section 5. We note that previous results [27; 28; 29] are proposed for the general non-i.i.d. case. Therefore, they may fail to give awesome guarantees in this special case. In Appendix C, we show that it is hard to derive a better bound than Theorem 3.1 by using the existing non-i.i.d. stability results directly.

*Remark.* **Stability of the learned distribution $\mathcal{D}_G(S)$.** $\mathcal{T}(m_S, m_G)$ in Theorem 3.1 reflects the stability of the learned distribution with regard to changing one data point in the training set received by the generative model. Our bound suggests that the more stable the model distribution is, the better performance can be achieved by GDA. As far as we know, though uniformly stability of some generative learning algorithms has been studied [40], the new notation $\mathcal{T}(m_S, m_G)$ emerging in our bound has not been studied yet.

*Remark.* **Selection of augmentation size.** We first consider the order of the upper bound with respect to $m_S$. Observing Theorem 3.1, we find that the distributions' divergence term can not be controlled by increasing $m_G$ while the remaining generalization error w.r.t. mixed distribution will vanish. We note that there exists a trade-off between the fast learning rate and augmentation consumption. Typically, the augmentation consumption is caused by additional sampling, training, and storage. When the order of the divergence term is smaller than that of the remains, increasing $m_G$ can induce a faster convergence. Otherwise, increasing $m_G$ can not lead to a faster convergence but a larger

consumption. Therefore, an efficient augmentation size $m^*_{G,\text{order}}$ with regard to the order of $m_S$ can be defined as follows:

$$m^*_{G,\text{order}} = \inf_{m_G} \{\text{generalization error w.r.t. mixed distribution} \lesssim \text{distributions' divergence}\}.$$

Furthermore, without considering the cost, the optimal augmentation number $m^*_G$ can be achieved by minimizing the upper bound directly. Unfortunately, it is difficult to calculate an explicit form of $m^*_{G,\text{order}}$ and $m^*_G$ here due to the ignorance of $\beta_{m_T}$ and $\mathcal{T}(m_S, m_G)$. We will discuss them more concretely in the specified cases.

*Remark.* **Sufficient conditions for GDA with (no) faster learning rate.** We still consider the order of the learning guarantee with respect to $m_S$ here. Let $m_G = m^*_{G,\text{order}}$, it can be found that divergence $\mathcal{D}_{\text{TV}}\left(\mathcal{D}, \mathcal{D}_G(S)\right)$ plays an important role in deciding whether GDA can enjoy a faster learning rate. Comparing Theorem 3.1 with Theorem 2.1 (without augmentation), we can conclude sufficient conditions as follows.

*Corollary* 3.1. *Assume that the loss function $\ell$ is bounded by $M$, we have*

- *if $\mathcal{D}_{\text{TV}}\left(\mathcal{D}, \mathcal{D}_G(S)\right) = o\left(\max\left(\log(m)\beta_m, 1/\sqrt{m}\right)\right)$, then GDA enjoys a faster learning rate.*

- *if $\mathcal{D}_{\text{TV}}\left(\mathcal{D}, \mathcal{D}_G(S)\right) = \Omega\left(\max\left(\log(m)\beta_m, 1/\sqrt{m}\right)\right)$, then GDA can not enjoy a faster learning rate.*

Notably, as we will present in Section 3.2 and 3.3, though GDA can not enjoy a faster learning rate in both special case, it is possible to improve the generalization guarantee at a constant level when $m_S$ is small, which is important when awful overfitting happens.

## 3.2 Theoretical results on bGMM

The bGMM is a classical but non-trivial setting, which has been widely studied in literature [41; 42; 43]. In this section, we investigated it in the context of GDA. Simulations will be conducted in Section 4.1 to verify these results.

### 3.2.1 Setting of bGMM

In this part, we introduce the data distribution configuration in the bGMM, as well as the corresponding linear classifier and conditional generative model. Similar setups of distribution and classifier have been adopted by many previous works [44; 45; 46].

**Distribution setting.** We consider a binary task where $\mathcal{Y} = \{-1, 1\}$. Given a vector $\boldsymbol{\mu} \in \mathbb{R}^d(\|\boldsymbol{\mu}\|_2 = 1)$ and noise variance $\sigma^2 > 0$, we assume that the distribution satisfies $y \sim \text{uniform}\{-1, 1\}$ and $\mathbf{x} \mid y \sim \mathcal{N}(y\boldsymbol{\mu}, \sigma^2 I_d)$. Besides, similarly to [47], we assume that the distribution of $y$ is known, which is satisfied in conditional learning with labels.

**Simple linear classifier.** We consider a linear classifier parameterized by $\boldsymbol{\theta} \in \mathbb{R}^d$ in the form of prediction $\widehat{y} = \text{sign}(\boldsymbol{\theta}^\top \mathbf{x})$. Given $m$ samples, $\boldsymbol{\theta}$ is learned by performing ERM with respect to the negative log-likelihood loss function, that is,

$$l(\boldsymbol{\theta}, (\mathbf{x}, y)) = \frac{1}{2\sigma^2}(\mathbf{x} - y\boldsymbol{\theta})^\top (\mathbf{x} - y\boldsymbol{\theta}).$$

As a result, this learning algorithm will return $\widehat{\boldsymbol{\theta}} = \frac{1}{m} \sum_{i=1}^{m} y_i \mathbf{x}_i$, which satisfies $\mathbb{E}[\widehat{\boldsymbol{\theta}}] = \boldsymbol{\mu}$.

**Conditional generative model.** We consider a simple generative model parameterized by $\boldsymbol{\mu}_y, \sigma^2_k$, where $y \in \{-1, 1\}$ and $k \in [d]$. It learns the parameters of Gaussian mixture distribution directly. Given $m$ data points, let $m_y$ be the number of samples in class $y$, it returns

$$\widehat{\boldsymbol{\mu}}_y = \frac{\sum_{y_i=y} \mathbf{x}_i}{m_y}, \quad \widehat{\sigma}^2_k = \sum_y \frac{m_y}{m} \frac{\sum_{y_i=y}(x_{ik} - \widehat{\mu}_{yk})^2}{m_y - 1},$$

which are unbiased estimators of $\pm\boldsymbol{\mu}$ and $\sigma^2$, respectively. Based on the learned parameters, we can perform GDA by generating new samples from the distribution $y \sim \text{uniform}\{-1, 1\}$, $\mathbf{x} \mid y \sim \mathcal{N}(\widehat{\boldsymbol{\mu}}_y, \widehat{\Sigma})$, where $\widehat{\Sigma} = \text{diag}(\widehat{\sigma}^2_1, \ldots, \widehat{\sigma}^2_d)$.

### 3.2.2 Theoretical results

In this section, we establish the generalization bound for bGMM based on the general bound proposed in Theorem 3.1. To derive such a bound, the main task is to bound terms $M$, $\beta_{m_T}$, $\mathcal{D}_{\mathrm{TV}}\left(\mathcal{D}, \mathcal{D}_G(S)\right)$ and $\mathcal{T}(m_S, m_G)$ in Theorem 3.1. For $M$ (Lemma B.5) and $\beta_{m_T}$ (Lemma B.6), we mainly use the concentration property of the multivariate Gaussian variable (Lemma B.4). In addition, inspired by previous works on naïve Bayes [48], we bound $\mathcal{D}_{\mathrm{TV}}\left(\mathcal{D}, \mathcal{D}_G(S)\right)$ (Lemma B.7) by discussing the distance between the estimated parameters and the true parameters of bGMM. Besides, the concentration property of $\mathcal{T}(m_S, m_G)$ (Lemma B.9) can be induced by the preceding discussion. Finally, we can obtain the following results.

**Theorem 3.2** (Generalization bound for bGMM, proof in Appendix B.2). *Consider the setting introduced in Section 3.2.1. Given a set $S$ with $m_S$ i.i.d. samples from the bGMM distribution $\mathcal{D}$ and an augmented set $S_G$ with $m_G$ i.i.d. samples drawn from the learned Gaussian mixture distribution, then with high probability at least $1 - \delta$, it holds that*

$$|Gen\text{-}error| \lesssim \begin{cases} \frac{\log(m_S)}{\sqrt{m_S}} & \text{if fix } d \text{ and } m_G = 0, \\ \frac{\log^2(m_S)}{\sqrt{m_S}} & \text{if fix } d \text{ and } m_G = \Theta(m_S), \\ \frac{\log(m_S)}{\sqrt{m_S}} & \text{if fix } d \text{ and } m_G = m_{G,\text{order}}^*, \\ d & \text{if fix } m_S. \end{cases} \tag{2}$$

*Remark.* **Explicit upper bound of generalization error.** (19) in Appendix B.2 gives us an explicit form to predict the generalization error in the bGMM setting. In Section 4.1, we will see that (19) predicts the order and trend of true generalization error well, which verifies the correctness of the proposed learning guarantee in the bGMM setting.

*Remark.* **Negative learning rate of GDA.** Even though we estimate the sufficient statistics of the Gaussian mixture distribution ($\boldsymbol{\mu}$ and $\sigma^2$) directly in this special case, we can not hope to enjoy a better learning rate. Things could be worse when we model the distribution in reality (e.g., images, texts), which suggests that when original samples are abundant, further performing GDA based on existing generative models can not improve the generalization. Theorem 3.3 supports this viewpoint.

*Remark.* **Improvement at a constant level matters a lot when overfitting happens.** From (2) we know that when $m_S$ is small and $d$ is large, the curse of dimensionality happens, which leads to an awful generalization error. In this case, though GDA can only improve it at a constant level by controlling the generalization error w.r.t. mixed distribution, the effect is obvious due to the large scale of $d$. We note that it is challenging to obtain an explicit form of the constant-level improvement due to the complexity of the generalization bound. Therefore, we clarify this more clearly by comparing the cases where $m_G = 0$ (without GDA) and $m_G \to +\infty$ in Corollary B.1 of Appendix B.2.

### 3.3 Implications on deep generative models

Nowadays, data augmentation with deep generative models is widely used and received lots of attention. Therefore, benefiting from the recent advances in the generative adversarial network (GAN) [2; 49] and SGD [35; 50], we discuss implications of our theory on real problems, which will be verified by the empirical experiments in Section 4.2.

#### 3.3.1 Learning setup

We consider the general binary classification task in the deep learning era. In this part, we introduce the setup of data distribution, deep neural classifier, learning algorithm, and deep generative model.

**Distribution setting.** We assume that input space satisfies $\mathcal{X} \subseteq [0, 1]^d$, and our analysis can be easily extended to any bounded input space. This assumption generally holds in many practical problems, for example, image data satisfies $\mathcal{X} \subseteq [0, 255]^d$. Similarly to bGMM, we let $\mathcal{Y} = \{-1, 1\}$ and assume that the distribution of $y$ is known.

**Deep neural classifier.** We consider a general $L$-layer multi-layer perception (MLP) or convolutional neural network (CNN) $f(\mathbf{w}, \cdot) : \mathcal{Z} \to \mathbb{R}$, where $\mathbf{w}$ denotes its weights and $\mathbf{w}_l$ denotes the weights in the $l$-th layer. Its abstract architecture is consistent with that in [50], and details can be found

in Appendix A.1. In addition, we suppose the deep neural classifier satisfies smoothness and boundedness assumptions, which are adopted by many previous works [34; 35; 50; 51].

**Assumption 3.1** (Smoothness). *We assume that $f(\mathbf{w}, \cdot)$ is $\eta$-smooth with respect to $\mathbf{w}$, that is, $|\nabla f(\mathbf{w}_1, \cdot) - \nabla f(\mathbf{w}_2, \cdot)| \leq \eta \|\mathbf{w}_1 - \mathbf{w}_2\|_2$ for any $\mathbf{w}_1$ and $\mathbf{w}_2$.*

**Assumption 3.2** (Boundedness). *We assume that for all $l \in [L]$, there exists a constant $W_l$, which satisfies $\|\mathbf{w}_l\|_2 \leq W_l$.*

**Learning algorithm for the deep neural classifier.** The setting of the learning algorithm is conformed to the practice. We assume that the loss function is the binary cross-entropy loss $\ell(f, (\mathbf{x}, y)) = \log(1 + \exp(-yf(\mathbf{w}, \mathbf{x})))$ and it is optimized by SGD. For the $t$-th step, we set the step size as $\frac{c}{\eta t}$ for some positive constant $c$. Besides, we assume that the total iteration number $T = O(m_T)$. These configurations are adopted by past works on the stability of SGD [34; 35].

**Deep generative model.** We choose GAN as our deep generative model, which is parameterized by MLP. Its abstract architecture is the same as that in Theorem 19 of [49], and details are placed in Appendix A.2. Besides, due to the lack of conditional generative model theory, we make a naive approximation here by assuming that each category is learned by a GAN, respectively.

### 3.3.2 Theoretical results

Similarly to the bGMM setting, we establish a generalization bound for the deep learning setup. To reach this goal, we bound terms $M$, $\beta_{m_T}$, and $\mathcal{D}_{\mathrm{TV}}\big(\mathcal{D}, \mathcal{D}_G(S)\big)$ based on the recent results on GAN [49] and SGD [29; 50]. First, boundedness and Lipschitzness of classifier $f$ can be induced from Assumption 3.2 (Lemma B.10). Second, the boundedness of $f$ directly implies the upper bound for $M$ because the binary cross-entropy loss is 1-Lipschitz with respect to $f$. Third, by combining the Lipschitzness and smoothness of $f$, we can bound $\beta_{m_T}$ for SGD (Lemma B.11). Finally, $\mathcal{D}_{\mathrm{TV}}\big(\mathcal{D}, \mathcal{D}_G(S)\big)$ can be bounded by the result in [49] (Lemma B.12).

**Theorem 3.3** (Generalization bound for GAN, proof in Appendix B.3 ). *Consider the setup introduced in Section 3.3.1. Given a set $S$ with $m_S$ i.i.d. samples from any distribution $\mathcal{D}$ and an augmented set $S_G$ with $m_G$ i.i.d. examples sampled from the distribution $\mathcal{D}_G(S)$ learned by GANs, then for any fixed $\delta \in (0, 1)$, with probability at least $1 - \delta$, it holds that*

$$
\mathbb{E}|\textit{Gen-error}| \lesssim
\begin{cases}
\frac{1}{\sqrt{m_S}} & \textit{if fix } W, L, d, \textit{ let } m_G = 0, \\[2mm]
\max\left(\left(\frac{\log(m_S)}{m_S}\right)^{\frac{1}{4}}, \log m_S \mathcal{T}(m_S, m_G)\right) & \textit{if fix } W, L, d, \textit{ let } m_G = \Theta(m_S), \\[2mm]
\left(\frac{\log(m_S)}{m_S}\right)^{\frac{1}{4}} & \textit{if fix } W, L, d, \textit{ let } m_G = m^*_{G,\mathrm{order}}, \\[2mm]
dL^2 \left(\prod_{l=1}^{L} \|W_l\|_2\right)^2 & \textit{if fix } m_S.
\end{cases}
$$

*Remark.* **Slow learning rate with GDA.** Upper bounds in Theorem 3.3 show that when we perform GDA, the order with regard to $m_S$ strictly becomes worse. Therefore, it implies that when $m_S$ is large enough, it is hopeless to boost the performance obviously by augmenting the train set based on GANs. On the contrary, GDA may make the generalization worse.

*Remark.* **GDA matters a lot when overfitting happens.** From Theorem 3.3, we know that as the data dimension and model capacity become larger, the deep neural classifier trained with SGD becomes easier to overfit the train set and gain terrible generalization performance. In this case, a constant-level improvement of generalization caused by GDA will be significant. Similarly to the bGMM setting, we clarify the constant-level improvement by comparing the cases where $m_G = 0$ (without GDA) and $m_G \to +\infty$ in Corollary B.2 of Appendix B.3.

## 4 Experiments

In this section, we conduct experiments to verify the results in Section 3, which are two-folded:

- We conduct simulations in the setting of bGMM and validate the results in Theorem 3.2.
- We empirically study the effect of GDA on the real CIFAR-10 dataset [52], which supports our theoretical implications on GANs.

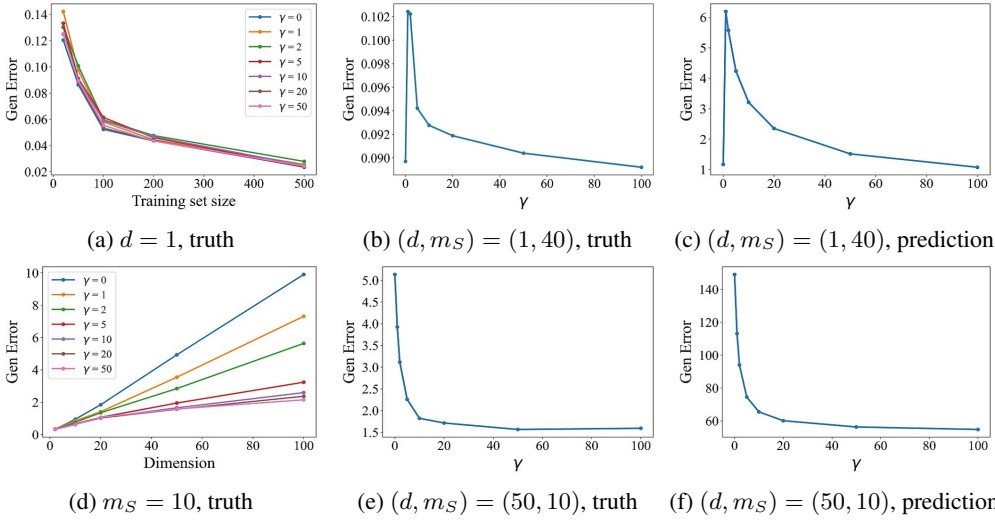

(a) $d = 1$, truth  (b) $(d, m_S) = (1, 40)$, truth  (c) $(d, m_S) = (1, 40)$, prediction

(d) $m_S = 10$, truth  (e) $(d, m_S) = (50, 10)$, truth  (f) $(d, m_S) = (50, 10)$, prediction

Figure 1: Simulations results on the bGMM setting. We do not integrate the truths and predictions due to the difference between their y-axis scale.

## 4.1 Simulations on bGMM

We let $\boldsymbol{\mu} = (1/\sqrt{d}, \dots, 1/\sqrt{d})^{\top}$ to satisfy $\|\boldsymbol{\mu}\|_2 = 1$, $\sigma^2 = 0.6^2$, and randomly generate 10,000 samples according to the Gaussian mixture distribution as the test set. We approximate the *Gen-error* by the gap between the training error and the test error. To eliminate randomness, we average over 1,000 random runs and report the mean results. We denote $\gamma = m_G/m_S$ in this section.

First, we investigate the case that data dimension $d$ is fixed. To verify the order is near to $O(1/\sqrt{m_S})$ ($\log m_S$ can be ignored with respect to $\sqrt{m_S}$), we fix $d = 1$, and change $m_S$ from 20 to 500. For each selected $m_S$, we adjust $\gamma$ from 0 to 50 to generate new samples in different levels. The result is presented in Figure 1a, which shows that the generalization error decreases in a near $O(1/\sqrt{m_S})$ order. Besides, generalization error without GDA is always (near) optimal, which empirically proves that GDA is ineffective when $m_S$ is large enough.

Second, we conduct simulations in the case that $m_S$ is fixed as a small constant. To verify the order is $O(d)$, we fix $m_S = 10$, and change $d$ from 2 to 100. For each selected $d$, we also adjust $\gamma$ from 0 to 50. The result is displayed in Figure 1d, which shows that the generalization error increases in a $O(d)$ order. In addition, when $d$ is large (e.g., 100) and the curse of dimensionality happens, generalization error with larger $\gamma$ is better by a big margin, which suggests that though GDA could only enhance it at a constant level, the effect is significant when overfitting occurs.

Third, we design experiments to validate whether the upper bound in Theorem 3.2 can predict the trend of generalization error well. Similarly to previous theoretical works (e.g. [53]), we find an approximation of (19) in Appendix B.2 as our prediction by replacing $\log(a/\delta)$ with $\log(a)$ if $a \neq 1$ else 1. We plot the ground truths and predictions in the case that $(d, m_S) = (1, 40)$ and $(50, 10)$, respectively. Results in Figure 1 show that our bound predicts the trend of generalization error well. Therefore, an approximation of the optimal augmentation size $m_G^*$ can be found by minimizing (19).

## 4.2 Empirical results on CIFAR-10

In this part, we conduct experiments on the real CIFAR-10 dataset with ResNets [54] and various deep generative models, including conditional DCGAN (cDCGAN) [55], StyleGAN2-ADA [56] and elucidating diffusion model (EDM) [30]. Details of experiments (e.g. motivation, model architecture, training) can be found in Appendix D.

To validate our theoretical implications in Section 3.3, we are supposed to discuss two cases, where one $m_S$ is small and the other $m_S$ is large. The two cases can be approximated by whether performing another data augmentation. We additionally use the standard data augmentation in [54] to approximate the case with large $m_S$. Then, for each selected ResNet and generative model, we set $m_G$ from 0 to

Table 1: Accuracy on the CIFAR-10 test set, where S.A. denotes standard augmentation.

| Generator | Classifier | S.A. | GDA ($m_G$) | | | | | |
|---|---|---|---|---|---|---|---|---|
| | | | 0 | 100k | 300k | 500k | 700k | 1M |
| cDCGAN [55] | ResNet18 | × | 85.76 | 86.8 | 87.83 | 87.59 | 87.52 | 86.47 |
| | | √ | 94.4 | 93.92 | 93.41 | 93.81 | 93.01 | 92.6 |
| | ResNet34 | × | 85 | 86.9 | 87.93 | 87.56 | 87.17 | 86.28 |
| | | √ | 94.59 | 94.83 | 94.21 | 93.64 | 93.69 | 93.18 |
| | ResNet50 | × | 82.85 | 87.49 | 88.59 | 86.67 | 86.3 | 85.2 |
| | | √ | 94.69 | 94.43 | 93.86 | 93.74 | 93.12 | 92.63 |
| StyleGAN2-ADA [56] | ResNet18 | × | 85.76 | 90.22 | 91.33 | 91.37 | 91.25 | 91.38 |
| | | √ | 94.4 | 94.68 | 94.46 | 94.4 | 94.11 | 94.12 |
| | ResNet34 | × | 85 | 90.24 | 91.23 | 91.45 | 91.56 | 90.91 |
| | | √ | 94.59 | 95.05 | 94.9 | 94.4 | 94.43 | 94.21 |
| | ResNet50 | × | 82.85 | 90.85 | 92.29 | 92.29 | 92.29 | 91.61 |
| | | √ | 94.69 | 94.74 | 95.04 | 94.56 | 94.76 | 94.28 |
| EDM [30] | ResNet18 | × | 85.76 | 92.8 | 94.87 | 95.43 | 96.24 | 96.28 |
| | | √ | 94.4 | 96.15 | 96.74 | 97.09 | 97.28 | 97.5 |
| | ResNet34 | × | 85 | 93.42 | 94.93 | 95.59 | 96.14 | 96.44 |
| | | √ | 94.59 | 96.47 | 96.96 | 97.36 | 97.53 | 97.51 |
| | ResNet50 | × | 82.85 | 93.29 | 95.29 | 95.95 | 96.1 | 96.64 |
| | | √ | 94.69 | 96.09 | 96.87 | 97.28 | 97.6 | 97.74 |

1M and record the accuracy of the trained classifier on the CIFAR-10 test set. Results are presented in Table 1. We further empirically verify our theory by estimating the generalization error directly. By definition, given a trained neural classifier, the generalization error of Theorem 3.3 can be estimated by the absolute gap between the mean cross-entropy loss on the training set (with generated data) and the mean cross-entropy loss on the test set. We place the results and analysis with this estimator at Table 3 in Appendix D.7. In the following, we interpret our experimental results.

**GANs improve the test performance of classifiers when overfitting occurs.** When standard augmentation is not used, ResNets trained on the train set consistently suffer from overfitting. However, this can be relieved by data augmentation based on GANs, though cDCGAN can not generate high-quality images. This phenomenon supports the implications from Theorem 3.3.

**We can not have an obvious improvement by using GANs when $m_S$ is approximately large.** When standard augmentation is used, deep neural classifiers trained on the CIFAR-10 dataset achieve non-trivial performance. In this case, GDA with cDCGAN always damages the generalization ability. Though we use StyleGAN2-ADA, which achieves state-of-the-art conditional image generation performance on the CIFAR-10 dataset, we can not boost the performance of classifiers obviously, and even consistently obtain worse test accuracy when $m_G$ is 500k or 1M.

**Diffusion probabilistic models are promising for GDA.** As diffusion models show their excellent ability on image generation, a natural question emerges: *are diffusion models more suitable for GDA?* We choose the EDM that achieves state-of-the-art FID scores as the generator. Table 1 in Appendix D.7 shows that EDM improves the test accuracy obviously, even though the standard augmentation has been utilized. This suggests that diffusion models enjoy $\mathcal{D}_{\mathrm{TV}}\left(\mathcal{D}, \mathcal{D}_G(S)\right)$ with a faster convergence rate than GANs, and shows the promise of diffusion models in GDA.

## 5 Related work

**Data augmentation practice and theory.** Data augmentation [57; 58] is a universal method to improve the generalization ability of deep neural networks in the case of insufficient training data. Classical data augmentation methods include geometric transformations [54], color space transformations [59], kernel filters [60], mixing images [61], random erasing [62], feature space augmentation [63], etc. There are also many theoretical works studying the effect of classical data augmentation methods from different perspectives [64; 65; 66; 67; 68].

With the advance of deep generative models, GDA becomes a novel and promising data augmentation technique. For example, [16] shows that augmenting the ImageNet training set [69] with samples from the conditional diffusion models significantly improves the classification accuracy. However, little work has investigated the theory of GDA. Both empirical success and theoretical opening encourage us to study the role of GDA.

**Algorithmic stability theory.** Classical results [25; 26] introduced detailedly in Section 2 has various extensions. Prominent work [34] focuses on the uniform stability of SGD and derive generalization bounds for it. [35] improves the results in [34] and obtains tight guarantees for the stability of SGD, which is used in Theorem 3.3.

Establishing stability bounds under non-i.i.d. settings has also received a surge of interest in recent years. A major line models the dependencies by mixing models [70; 71] and derives stability bounds with mixing coefficients [27; 28; 72]. However, it is usually difficult to estimate the mixing coefficients quantitatively. To avoid this problem, another line qualitatively models the dependencies by graphs. Recently, [29] derive a general stability bound for dependent settings characterized by forest complexity of the dependency graph. However, it is hard to use these techniques to derive a better bound than Theorem 3.1 for GDA, which is discussed detailedly in Appendix C.

**Convergence of deep generative models.** In addition to the bound for $\mathcal{D}_{\mathrm{TV}}\left(\mathcal{D}, \mathcal{D}_G(S)\right)$ with respect to GANs [49] we used in Theorem 3.3, there are attempts to derive such a bound for diffusion models [73; 74; 75; 76]. Informally, they mainly assume that estimation error of score function is bounded, then with an appropriate choice of step size and iteration number, diffusion models output a distribution which is close to the true distribution. However, it is still unclear how to derive learning guarantees with respect to the train set size $m_S$ directly. Once such learning guarantees are established, we can directly analyze the effect of GDA with diffusion models by Theorem 3.1.

## 6 Impacts and limitations

This paper is mainly a theoretical work and a first step towards understanding the GDA, and it can give some insights to the practice. Theorem 3.1 implies that improving the distribution approximation performance of the generative models is important for the GDA, which motivates people to design better generative models. Besides, it shows that stabilizing the training of the generative models can bring benefits to the GDA, which motivates us to improve the stability of the training of generative models (e.g. GAN). Furthermore, it implies that if we can estimate terms in the bound, then the optimal augmentation size can be approximated. With the emergence of more advanced theory, our results can be extended to other settings (e.g., diffusion models, self-supervised learning) and give more guidance to the practice. One limitation is that our results do not enjoy tightness guarantees, though they benefit from the same nearly optimal guarantee shown by [26] when $m_G = 0$. The derivation of lower bounds can be left to future work.

## 7 Conclusion

In this paper, we attempt to understand modern GDA techniques. To realize this goal, we first establish a general algorithmic stability bound in this non-i.i.d. setting. It suggests that GDA enjoys a faster learning rate when the divergence term $\mathcal{D}_{\mathrm{TV}}\left(\mathcal{D}, \mathcal{D}_G(S)\right) = o(\max\left(\log(m)\beta_m, 1/\sqrt{m}\right))$. Second, We specify the learning guarantee to the bGMM and GANs settings. Theoretical results show that, in both cases, though GDA can not enjoy a faster learning rate, it is effective when terrible overfitting happens, which suggests its promise in learning with limited data. Finally, experimental results support our theoretical conclusions and further show the promise of diffusion models in GDA.

## Acknowledgement

This work was supported by NSF of China (Nos. 62076145, 62206159); Beijing Outstanding Young Scientist Program (No. BJJWZYJH012019100020098); Shandong Provincial Natural Science Foundation (No. ZR2022QF117); Major Innovation & Planning Interdisciplinary Platform for the "Double-First Class" Initiative, Renmin University of China; the Fundamental Research Funds for the Central Universities, and the Research Funds of Renmin University of China (No. 22XNKJ13); the Fundamental Research Funds of Shandong University. C. Li was also sponsored by Beijing Nova Program (No. 20220484044).

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
