# Contents of Appendix

# Appendix A  Architectures of deep neural networks in Section 3.3.1

## A.1  Architecture of deep neural classifier in Section 3.3

We consider a general class of neural networks as what is introduced in [50], which includes widely used MLPs and CNNs. We define a deep neural network with $L_C$ convolutional layers followed by $L - L_C - 1$ fully-connected layers as follows:

$$f(\mathbf{x}; \mathbf{w}) = \sum_{k=1}^{m} a_k z_{(L-1),k}$$
$$\mathbf{z}_l = \sigma\left(\mathbf{A}_l^\top \mathbf{z}_{(l-1)}\right), l \in [L-1] - [L_C],$$
$$\mathbf{z}_l = \text{pool}\left(\mathbf{y}_l\right), l \in [L_C],$$
$$\mathbf{y}_l = \sigma\left(\mathbf{w}_l * \mathbf{z}_{(l-1)}\right), l \in [L_C],$$
$$\mathbf{z}_0 = \mathbf{x}$$

where $m$ is the demension of $\mathbf{z}_{(L-1)}$, $\sigma(z)$ is the ReLU function $\max\{z, 0\}$, * is the convolutional operation, and $\text{pool}(\cdot)$ is the average pooling operation. When $L_C = 0$, this is an MLP. For output layer $l = L$, let $\mathbf{w}_L := (a_1, \cdots, a_m)^\top$. For fully-connected layer $l \in [L-1] - [L_C]$, we let $\mathbf{w}_l := \text{vector}(\mathbf{A}_l)$. For convolution layer $l \in [L_C]$, we consider the structure Convolution $\to$ ReLU $\to$ Pooling, and denotes the weights as $\mathbf{w}_l$.

## A.2  Architecture of GAN in Section 3.3

**The abstract form of GAN.** The architecture of GAN in Theorem 3.3 is consistent with that in Theorem 19, [49]. We denote by $\mathcal{F} = \{f_{\boldsymbol{\omega}}(\mathbf{x}) : \mathbb{R}^d \to \mathbb{R}\}$ the discriminator function space. Besides, we let $\mathcal{G} = \{g_{\boldsymbol{\theta}}(\mathbf{z}) : \mathbb{R}^d \to \mathbb{R}^d\}$ be the generator function space. The generator receives $\mathbf{z} \sim \text{unif}[0, 1]^d$ as the random input. In reality, we estimate the parameters of GAN as

$$\widehat{\boldsymbol{\theta}}_{m,n} \in \arg\min_{\boldsymbol{\theta}: g_{\boldsymbol{\theta}} \in \mathcal{G}} \max_{\boldsymbol{\omega}: f_{\boldsymbol{\omega}} \in \mathcal{F}} \left\{\widehat{\mathbb{E}}_n f_{\boldsymbol{\omega}}\left(g_{\boldsymbol{\theta}}(Z)\right) - \widehat{\mathbb{E}}_m f_{\boldsymbol{\omega}}(X)\right\},$$

where $n$ and $m$ denote the number of simulated and target distribution samples, respectively. We just let $m = n$ in this paper.

**The architecture of the generator network.** The generator $g_{\boldsymbol{\theta}}$ is parametrized by a MLP:

$$\mathbf{h}_0 = \mathbf{z},$$
$$\mathbf{h}_l = \sigma_a\left(\mathbf{W}_l \mathbf{h}_{l-1} + \mathbf{b}_l\right), 0 < l < L$$
$$\mathbf{x} = \mathbf{W}_L \mathbf{h}_{L-1} + \mathbf{b}_L,$$

where $h_l$ denotes the hidden units in the $l$-th layer, and $\mathbf{x}$ is the final output of the MLP. The activation is leaky ReLU [77].

$$\sigma_a(t) = \max\{t, at\}, \text{ for some fixed } 0 < a \leq 1$$

The space for the generator weights is denoted by

$$\Theta(d, L) := \left\{\boldsymbol{\theta} = \left(\mathbf{W}_l \in \mathbb{R}^{d \times d}, \mathbf{b}_l \in \mathbb{R}^d, 1 \leq l \leq L\right) \mid \text{rank}\left(\mathbf{W}_l\right) = d, \forall 1 \leq l \leq L\right\}.$$

Note that the $\mathbf{W}_l$ is required to be full rank so that the generator transformation $g_{\theta}$ is invertible. The generator has the capacity to express complex distributions

**The architecture of the discriminator network.** We consider a discriminator network which includes feed-forward neural networks $f_{\boldsymbol{\omega}}$ that satisfies

$$\mathbf{h}_1 = \sigma_{1/a} \left( \mathbf{V}_1 \mathbf{x} + \mathbf{c}_1 \right)$$

$$\cdots$$

$$\mathbf{h}_{L-1} = \sigma_{1/a} \left( \mathbf{V}_{L-1} \mathbf{h}_{L-2} + \mathbf{c}_{L-1} \right)$$

$$q_{\boldsymbol{\omega}}(\mathbf{x}) := \sum_{j=1}^{L-1} \sum_{i=1}^{d} \log(1/a) 1_{h_{ji} \leq 0} + c_L.$$

The parameter space is defined as

$$\Omega(d, L) := \left\{ \boldsymbol{\omega} = \left( \mathbf{V}_l \in \mathbb{R}^{d \times d}, \mathbf{c}_l \in \mathbb{R}^d, c_L \in \mathbb{R}, 1 \leq l \leq L-1 \right) \mid \operatorname{rank}(\mathbf{V}_l) = d, \forall 1 \leq l \leq L-1 \right\}.$$

Finally, the discriminator parameterized by $\boldsymbol{\omega} = (\boldsymbol{\omega}_1, \boldsymbol{\omega}_2)$, where $\boldsymbol{\omega}_1, \boldsymbol{\omega}_2 \in \Omega(d, L)$, is defined as

$$f_{\boldsymbol{\omega}}(\mathbf{x}) = q_{\boldsymbol{\omega}_1}(\mathbf{x}) - q_{\boldsymbol{\omega}_2}(\mathbf{x}).$$

## Appendix B   Proofs

### B.1   Proof of Theorem 3.1

*Proof.* We first list some moment inequalities which are important to this proof.

**Lemma B.1** (Lemma 1, [26])**.** *If $\|Y\|_p \leq \sqrt{p}a + pb$ for any $p \geq 1$, then for any $\delta \in (0,1)$, with probability at least $1 - \delta$,*

$$|Y| \leq e \left( a \sqrt{\log\left(\frac{e}{\delta}\right)} + b \log\left(\frac{e}{\delta}\right) \right).$$

**Lemma B.2** (Lemma 2, [26])**.** *Consider a function $f$ of independent random variables $X_1, \ldots, X_n$ where $X_i \in \mathcal{X}$. Suppose that for any $i = 1, \ldots, n$ and any $x_1, \ldots, x_n, x_i' \in \mathcal{X}$ it holds that*

$$|f(x_1, \ldots, x_n) - f(x_1, \ldots, x_{i-1}, x_i', x_{i+1}, \ldots, x_n)| \leq \beta. \tag{3}$$

*Then, we have for any $p \geq 2$,*

$$\|f(X_1, \ldots, X_n) - \mathbb{E}f(X_1, \ldots, X_n)\|_p \leq 2\sqrt{np}\beta.$$

**Lemma B.3** (Theorem 4, [26])**.** *Let $\mathbf{Z} = (Z_1, \ldots, Z_n)$ be a vector of independent random variables each taking values in $\mathcal{Z}$, and let $g_1, \ldots, g_n$ be some functions $g_i : \mathcal{Z}^n \to \mathbb{R}$ such that the following holds for any $i \in [n]$:*

- *$\left| \mathbb{E}[g_i(\mathbf{Z})|Z_i] \right| \leq M$,*

- *$\mathbb{E}[g_i(\mathbf{Z})|\mathbf{Z}^{\setminus i}] = 0$,*

- *$g_i$ has a bounded difference $\beta$ with respect to all variables except the $i$-th variable, that is, for all $j \neq i$, $\mathbf{Z} = (Z_1, \ldots, Z_n)$ and $\mathbf{Z}^j = (Z_1, \ldots, Z_j', \ldots, Z_n) \in \mathbb{R}^n$, we have $\left| g_i(\mathbf{Z}) - g_i(\mathbf{Z}^j) \right| \leq \beta$.*

*Then, for any $p \geq 2$,*

$$\left\| \sum_{i=1}^{n} g_i(\mathbf{Z}) \right\|_p \leq 12\sqrt{2}pn\beta \log n + 4M\sqrt{pn}.$$

Now, we are ready to prove Theorem 3.1. Formally, we need to bound *Gen-error* $= |\mathcal{R}_{\mathcal{D}}(\mathcal{A}(\widetilde{S})) - \widehat{\mathcal{R}}_{\widetilde{S}}(\mathcal{A}(\widetilde{S}))|$. Recall that $\widetilde{\mathcal{D}}(S)$ has been defined as the mixed distribution after augmentation, to derive such a bound, we first decomposed *Gen-error* as

$$|\textit{Gen-error}| \leq \underbrace{\left| \mathcal{R}_{\mathcal{D}}(\mathcal{A}(\widetilde{S})) - \mathcal{R}_{\widetilde{\mathcal{D}}(S)}(\mathcal{A}(\widetilde{S})) \right|}_{\text{Distributions' divergence}} + \underbrace{\left| \mathcal{R}_{\widetilde{\mathcal{D}}(S)}(\mathcal{A}(\widetilde{S})) - \widehat{\mathcal{R}}_{\widetilde{S}}(\mathcal{A}(\widetilde{S})) \right|}_{\text{Generaliztion error w.r.t. mixed distribution, } \Phi(S, S_G)}.$$

The distributions' divergence term in the right hand can be bounded by the divergence (e.g., $\mathcal{D}_{\mathrm{TV}}, \mathcal{D}_{\mathrm{KL}}$) between augmented distribution $\widetilde{\mathcal{D}}(S)$ and the true distribution $\mathcal{D}$. It is heavily dependent on the ability of the chosen generative model. It can be bounded as follows.

$$
\begin{aligned}
\left|\mathcal{R}_{\mathcal{D}}(\mathcal{A}(\widetilde{S})) - \mathcal{R}_{\widetilde{\mathcal{D}}(S)}(\mathcal{A}(\widetilde{S}))\right| &= \frac{m_G}{m_T}\left|\mathcal{R}_{\mathcal{D}}(\mathcal{A}(\widetilde{S})) - \mathcal{R}_{\mathcal{D}_G(S)}(\mathcal{A}(\widetilde{S}))\right| \\
&= \frac{m_G}{m_T}\left|\int_{\mathbf{z}} \ell(\mathcal{A}(\widetilde{S}), \mathbf{z})\left(\mathbb{P}_{\mathcal{D}}(\mathbf{z}) - \mathbb{P}_{\mathcal{D}_G(S)}(\mathbf{z})\right) d\mathbf{z}\right| \\
&\leq \frac{m_G}{m_T}\int_{\mathbf{z}}\left|\ell(\mathcal{A}(\widetilde{S}), \mathbf{z})\left(\mathbb{P}_{\mathcal{D}}(\mathbf{z}) - \mathbb{P}_{\mathcal{D}_G(S)}(\mathbf{z})\right)\right| d\mathbf{z} \\
&\leq \frac{m_G}{m_T}M\int_{\mathbf{z}}\left|\mathbb{P}_{\mathcal{D}}(\mathbf{z}) - \mathbb{P}_{\mathcal{D}_G(S)}(\mathbf{z})\right| d\mathbf{z} \\
&\lesssim \frac{m_G}{m_T}M\mathcal{D}_{\mathrm{TV}}\left(\mathcal{D}, \mathcal{D}_G(S)\right).
\end{aligned}
$$

For the second term $\Phi(S, S_G)$, we note that classical stability bounds (e.g. Theorem 2.1) can not be used directly, because points in $\widetilde{S}$ are drawn non-i.i.d.. In contrast, a core property of $\widetilde{S}$ is that $S$ satisfies i.i.d. assumption, and $S_G$ satisfies conditional i.i.d. assumption when $S$ is fixed. Inspired by this property, we furthermore decomposed this term and utilized sharp moment inequalities [39; 26] to obtain an upper bound. Similarly to [26], we bound the $L_p$ norm of $m_T\Phi(S, S_G)$, and then derive a concentration bound. We can write

$$
\begin{aligned}
\left\|m_T\Phi(S, S_G)\right\|_p &= \left\|m_T\left(\mathcal{R}_{\widetilde{\mathcal{D}}(S)}(\mathcal{A}(\widetilde{S})) - \widehat{\mathcal{R}}_{\widetilde{S}}(\mathcal{A}(\widetilde{S}))\right)\right\|_p \\
&= \left\|m_S\mathcal{R}_{\mathcal{D}}(\mathcal{A}(\widetilde{S})) + m_G\mathcal{R}_{\mathcal{D}_G(S)}(\mathcal{A}(\widetilde{S})) - \sum_{\mathbf{z}_i \in S}\ell(\mathcal{A}(\widetilde{S}), \mathbf{z}_i) - \sum_{\mathbf{z}_i \in S_G}\ell(\mathcal{A}(\widetilde{S}), \mathbf{z}_i)\right\|_p \\
&\leq \underbrace{\left\|m_S\mathcal{R}_{\mathcal{D}}(\mathcal{A}(\widetilde{S})) - \sum_{i=1}^{m_S}\ell(\mathcal{A}(\widetilde{S}), \mathbf{z}_i)\right\|_p}_{\left\|\Phi_1(S, S_G)\right\|_p} + \underbrace{\left\|m_G\mathcal{R}_{\mathcal{D}_G(S)}(\mathcal{A}(\widetilde{S})) - \sum_{i=1}^{m_G}\ell(\mathcal{A}(\widetilde{S}), \mathbf{z}_i^G)\right\|_p}_{\left\|\Phi_2(S, S_G)\right\|_p}.
\end{aligned}
$$

We will bound $\left\|\Phi_1(S, S_G)\right\|_p$ and $\left\|\Phi_2(S, S_G)\right\|_p$ respectively. We note that for any function $f(S)$, if we have an bound $\|f\|_p(S_V) \leq C$ for some $S_V \subseteq S$, then we have

$$
\|f\|_p = (\mathbb{E}\mathbb{E}[|f|^p|S_V])^{1/p} \leq (\mathbb{E}[C^p])^{1/p} \leq C. \tag{4}
$$

Fix $S$, then data in $S_G$ are independent. We use this property and lemma B.3 to bound $\left\|\Phi_2\right\|_p(S)$. We introduce functions $f_i(S_G)$ which play the same role as $g_i$s in Lemma B.3, as

$$
f_i(S_G) = \mathbb{E}_{\mathbf{z}_i' \sim \mathcal{D}_G(S)}\left[\mathbb{E}_{\mathbf{z} \sim \mathcal{D}_G(S)}\ell(\mathcal{A}(S \cup S_G^i), \mathbf{z}) - \ell(\mathcal{A}(S \cup S_G^i), \mathbf{z}_i^G)\right],
$$

where $\mathbf{z}_i^G$ is the $i$-th data in $S_G$, and $S_G^i$ obtained by replacing $\mathbf{z}_i^G$ by $\mathbf{z}_i'$. We note that $|f_i| \leq M$, $\mathbb{E}[f_i|S_G^{\setminus i}] = 0$ and $f_i$ has a bounded difference $2\beta_{m_T}$ with respect to all variables except the $i$-th variable, which can be proved as follows.

$$
\begin{aligned}
|f_i| &= \left|\mathbb{E}_{\mathbf{z}_i' \sim \mathcal{D}_G(S)}\left[\mathbb{E}_{\mathbf{z} \sim \mathcal{D}_G(S)}\ell(\mathcal{A}(S \cup S_G^i), \mathbf{z}) - \ell(\mathcal{A}(S \cup S_G^i), \mathbf{z}_i^G)\right]\right| \\
&= \left|\mathbb{E}_{\mathbf{z}_i' \sim \mathcal{D}_G(S)}\mathbb{E}_{\mathbf{z} \sim \mathcal{D}_G(S)}\left[\ell(\mathcal{A}(S \cup S_G^i), \mathbf{z}) - \ell(\mathcal{A}(S \cup S_G^i), \mathbf{z}_i^G)\right]\right|
\end{aligned}
$$

$$\leq \mathbb{E}_{\mathbf{z}'_i \sim \mathcal{D}_G(S)} \mathbb{E}_{\mathbf{z} \sim \mathcal{D}_G(S)} \left| \ell(\mathcal{A}(S \cup S_G^i), \mathbf{z}) - \ell(\mathcal{A}(S \cup S_G^i), \mathbf{z}_i^G) \right|$$

$$\leq \mathbb{E}_{\mathbf{z}'_i \sim \mathcal{D}_G(S)} \mathbb{E}_{\mathbf{z} \sim \mathcal{D}_G(S)}[M] = M,$$

$$\mathbb{E}[f_i | S_G^{\backslash i}] = \mathbb{E}_{\mathbf{z}_i^G \sim \mathcal{D}_G(S)} \left[ \mathbb{E}_{\mathbf{z}'_i \sim \mathcal{D}_G(S)} \left[ \mathbb{E}_{\mathbf{z} \sim \mathcal{D}_G(S)} \ell(\mathcal{A}(S \cup S_G^i), \mathbf{z}) - \ell(\mathcal{A}(S \cup S_G^i), \mathbf{z}_i^G) \right] | S_G^{\backslash i} \right]$$

$$= \mathbb{E}_{\mathbf{z}'_i \sim \mathcal{D}_G(S)} \left[ \left[ \mathbb{E}_{\mathbf{z} \sim \mathcal{D}_G(S)} \ell(\mathcal{A}(S \cup S_G^i), \mathbf{z}) - \mathbb{E}_{\mathbf{z}_i^G \sim \mathcal{D}_G(S)} \ell(\mathcal{A}(S \cup S_G^i), \mathbf{z}_i^G) \right] | S_G^{\backslash i} \right]$$

$$= \mathbb{E}_{\mathbf{z}'_i \sim \mathcal{D}_G(S)} \left[ 0 | S_G^{\backslash i} \right] = 0,$$

$$\left| f_i(S_G) - f_i(S_G^j) \right| = \left| \mathbb{E}_{\mathbf{z}'_i \sim \mathcal{D}_G(S)} \left[ \mathbb{E}_{\mathbf{z} \sim \mathcal{D}_G(S)} \ell(\mathcal{A}(S \cup S_G^i), \mathbf{z}) - \ell(\mathcal{A}(S \cup S_G^i), \mathbf{z}_i^G) \right] \right.$$

$$\left. - \mathbb{E}_{\mathbf{z}'_i \sim \mathcal{D}_G(S)} \left[ \mathbb{E}_{\mathbf{z} \sim \mathcal{D}_G(S)} \ell(\mathcal{A}(S \cup (S_G^j)^i), \mathbf{z}) - \ell(\mathcal{A}(S \cup (S_G^j)^i, \mathbf{z}_i^G) \right] \right|$$

$$= \left| \mathbb{E}_{\mathbf{z}'_i \sim \mathcal{D}_G(S)} \left[ \mathbb{E}_{\mathbf{z} \sim \mathcal{D}_G(S)} \ell(\mathcal{A}(S \cup S_G^i), \mathbf{z}) - \ell(\mathcal{A}(S \cup S_G^i), \mathbf{z}_i^G) \right. \right.$$

$$\left. \left. - \mathbb{E}_{\mathbf{z} \sim \mathcal{D}_G(S)} \ell(\mathcal{A}(S \cup (S_G^j)^i), \mathbf{z}) + \ell(\mathcal{A}(S \cup (S_G^j)^i, \mathbf{z}_i^G) \right] \right|$$

$$\leq \left| \mathbb{E}_{\mathbf{z}'_i \sim \mathcal{D}_G(S)} \mathbb{E}_{\mathbf{z} \sim \mathcal{D}_G(S)} \left[ \ell(\mathcal{A}(S \cup S_G^i), \mathbf{z}) - \ell(\mathcal{A}(S \cup (S_G^j)^i), \mathbf{z}) \right] \right|$$

$$+ \left| \mathbb{E}_{\mathbf{z}'_i \sim \mathcal{D}_G(S)} \left[ \ell(\mathcal{A}(S \cup S_G^i), \mathbf{z}_i^G) - \ell(\mathcal{A}(S \cup (S_G^j)^i), \mathbf{z}_i^G) \right] \right|$$

$$\leq \mathbb{E}_{\mathbf{z}'_i \sim \mathcal{D}_G(S)} \mathbb{E}_{\mathbf{z} \sim \mathcal{D}_G(S)} \left| \ell(\mathcal{A}(S \cup S_G^i), \mathbf{z}) - \ell(\mathcal{A}(S \cup (S_G^j)^i), \mathbf{z}) \right|$$

$$+ \mathbb{E}_{\mathbf{z}'_i \sim \mathcal{D}_G(S)} \left| \ell(\mathcal{A}(S \cup S_G^i), \mathbf{z}_i^G) - \ell(\mathcal{A}(S \cup (S_G^j)^i), \mathbf{z}_i^G) \right|$$

$$\leq \beta_{m_T} + \beta_{m_T} = 2\beta_{m_T}.$$

Therefore, for any fixed $S$, by Lemma B.3, for any $p \geq 2$, we have

$$\left\| \sum_{i=1}^{m_G} f_i(S_G) \right\|_p \lesssim p m_G \beta_{m_T} \log m_G + M \sqrt{p m_G}. \tag{5}$$

We note the gap between $\Phi_2$ and $\sum_{i=1}^{m_G} f_i$ is small, then for any fixed $S$, we can bound $\|\Phi_2\|_p (S)$ by (5) as follows.

$$\|\Phi_2\|_p (S) = \left\| m_G \mathcal{R}_{\mathcal{D}_G(S)}(\mathcal{A}(S \cup S_G)) - \sum_{i=1}^{m_G} \ell(\mathcal{A}(S \cup S_G), \mathbf{z}_i^G) \right\|_p$$

$$= \left\| \sum_{i=1}^{m_G} \left( \mathbb{E}_{\mathbf{z} \sim \mathcal{D}_G(S)} \ell(\mathcal{A}(S \cup S_G), \mathbf{z}) - \ell(\mathcal{A}(S \cup S_G), \mathbf{z}_i^G) \right) \right\|_p$$

$$\leq \left\| \sum_{i=1}^{m_G} \left( \mathbb{E}_{\mathbf{z}'_i \sim \mathcal{D}_G(S)} \left[ \mathbb{E}_{\mathbf{z} \sim \mathcal{D}_G(S)} \ell(\mathcal{A}(S \cup S_G^i), \mathbf{z}) - \ell(\mathcal{A}(S \cup S_G^i), \mathbf{z}_i^G) \right] \right) \right\|_p + \left\| 2 m_G \beta_{m_T} \right\|_p$$

$$= \left\| \sum_{i=1}^{m_G} f_i(S_G) \right\|_p + \left\| 2m_G \beta_{m_T} \right\|_p$$
$$\lesssim pm_G \beta_{m_T} \log m_G + M\sqrt{pm_G} + 2m_G \beta_{m_T}$$
$$\lesssim pm_G \beta_{m_T} \log m_G + M\sqrt{pm_G}.$$

Therefore, by using (4), we have

$$\left\| \Phi_2(S, S_G) \right\|_p \lesssim pm_G \beta_{m_T} \log m_G + M\sqrt{pm_G}. \tag{6}$$

Now, we use a similar idea to bound $\left\| \Phi_1(S, S_G) \right\|_p$. We decompose $\left\| \Phi_1(S, S_G) \right\|_p$ as the following.

$$\left\| \Phi_1(S, S_G) \right\|_p = \left\| \Phi_1 - \mathbb{E}_{S_G \sim \mathcal{D}_G^{m_G}(S)} \Phi_1 + \mathbb{E}_{S_G \sim \mathcal{D}_G^{m_G}(S)} \Phi_1 \right\|_p$$
$$\leq \underbrace{\left\| \Phi_1 - \mathbb{E}_{S_G \sim \mathcal{D}_G^{m_G}(S)} \Phi_1 \right\|_p}_{\Delta_1} + \underbrace{\left\| \mathbb{E}_{S_G \sim \mathcal{D}_G^{m_G}(S)} \Phi_1 \right\|}_{\Delta_2},$$

We then bound each term and obtain a bound for $\left\| \Phi_1(S, S_G) \right\|_p$. We note that $\Delta_1$ can be bounded by using Lemma B.2 and $\Delta_2$ can be bounded by using Lemma B.3.

To bound $\Delta_1$, we first fix $S$ and bound $\left\| \Phi_1 - \mathbb{E}_{S_G \sim \mathcal{D}_G^{m_G}(S)} \Phi_1 \right\|_p (S)$. We use the conditional independence property of $S_G$ again. To use Lemma B.2, we need to prove that $\Phi_1$ has the bounded difference with respect to $S_G$ when $S$ is fixed. We can write

$$\left| \Phi_1(S, S_G) - \Phi_1(S, S_G^i) \right|$$
$$= \left| m_S \mathcal{R}_\mathcal{D}(\mathcal{A}(S \cup S_G)) - \sum_{i=1}^{m_S} \ell(\mathcal{A}(S \cup S_G), \mathbf{z}_i) - m_S \mathcal{R}_\mathcal{D}(\mathcal{A}(S \cup S_G^i)) + \sum_{i=1}^{m_S} \ell(\mathcal{A}(S \cup S_G^i), \mathbf{z}_i) \right|$$
$$\leq m_S \left| \mathcal{R}_\mathcal{D}(\mathcal{A}(S \cup S_G)) - \mathcal{R}_\mathcal{D}(\mathcal{A}(S \cup S_G^i)) \right| + \sum_{i=1}^{m_S} \left| \ell(\mathcal{A}(S \cup S_G), \mathbf{z}_i) - \ell(\mathcal{A}(S \cup S_G^i), \mathbf{z}_i) \right|$$
$$\leq m_S \beta_{m_T} + m_S \beta_{m_T} = 2m_S \beta_{m_T}.$$

Thus, by Lemma B.2, we have

$$\Delta_1 \leq 4\sqrt{m_G p} m_S \beta_{m_T} \lesssim \sqrt{m_G p} m_S \beta_{m_T}. \tag{7}$$

We now construct some functions and use Lemma B.3 again to bound $\Delta_2$. We define $h_i(S)$ which play the same role as $g_i$s in Lemma B.3, as

$$h_i(S) = \mathbb{E}_{\mathbf{z}_i' \sim \mathcal{D}} \mathbb{E}_{S_G \sim \mathcal{D}_G^{m_G}(S^i)} \left[ \mathbb{E}_{\mathbf{z} \sim \mathcal{D}} \ell(\mathcal{A}(S^i \cup S_G), \mathbf{z}) - \ell(\mathcal{A}(S^i \cup S_G), \mathbf{z}_i) \right],$$

where $\mathbf{z}_i$ is the $i$-th data in $S$, and $S^i$ obtained by replacing $\mathbf{z}_i$ by $\mathbf{z}_i'$. We note that $|h_i| \leq M$, $\mathbb{E}[h_i | S^{\setminus i}] = 0$ and $h_i$ has a bounded difference $2\beta_{m_T} + 2M\mathcal{T}(m_S, m_G)$ with respect to all variables except the $i$-th variable, where $\mathcal{T}(m_S, m_G) = \sup_i \mathcal{D}_{\text{TV}} \left( \mathcal{D}_G^{m_G}(S), \mathcal{D}_G^{m_G}(S^i) \right)$. These can be proved as follows.

$$|h_i| = \left| \mathbb{E}_{\mathbf{z}_i' \sim \mathcal{D}} \mathbb{E}_{S_G \sim \mathcal{D}_G^{m_G}(S^i)} \left[ \mathbb{E}_{\mathbf{z} \sim \mathcal{D}} \ell(\mathcal{A}(S^i \cup S_G), \mathbf{z}) - \ell(\mathcal{A}(S^i \cup S_G), \mathbf{z}_i) \right] \right|$$
$$= \left| \mathbb{E}_{\mathbf{z}_i' \sim \mathcal{D}} \mathbb{E}_{S_G \sim \mathcal{D}_G^{m_G}(S^i)} \mathbb{E}_{\mathbf{z} \sim \mathcal{D}} \left[ \ell(\mathcal{A}(S^i \cup S_G), \mathbf{z}) - \ell(\mathcal{A}(S^i \cup S_G), \mathbf{z}_i) \right] \right|$$

$$= \mathbb{E}_{\mathbf{z}'_i \sim \mathcal{D}} \mathbb{E}_{S_G \sim \mathcal{D}_G^{m_G}(S^i)} \mathbb{E}_{\mathbf{z} \sim \mathcal{D}} \left| \ell(\mathcal{A}(S^i \cup S_G), \mathbf{z}) - \ell(\mathcal{A}(S^i \cup S_G), \mathbf{z}_i) \right|$$
$$\leq M,$$

$$\mathbb{E}[h_i | S^{\backslash i}] = \mathbb{E}_{\mathbf{z}_i \sim \mathcal{D}} \left[ \mathbb{E}_{\mathbf{z}'_i \sim \mathcal{D}} \mathbb{E}_{S_G \sim \mathcal{D}_G^{m_G}(S^i)} \left[ \mathbb{E}_{\mathbf{z} \sim \mathcal{D}} \ell(\mathcal{A}(S^i \cup S_G), \mathbf{z}) - \ell(\mathcal{A}(S^i \cup S_G), \mathbf{z}_i) \right] | S^{\backslash i} \right]$$
$$= \mathbb{E}_{\mathbf{z}'_i \sim \mathcal{D}} \mathbb{E}_{S_G \sim \mathcal{D}_G^{m_G}(S^i)} \left[ \left[ \mathbb{E}_{\mathbf{z} \sim \mathcal{D}} \ell(\mathcal{A}(S^i \cup S_G), \mathbf{z}) - \mathbb{E}_{\mathbf{z}_i \sim \mathcal{D}} \ell(\mathcal{A}(S^i \cup S_G), \mathbf{z}_i) \right] | S^{\backslash i} \right]$$
$$= 0,$$

$$\left| h_i(S) - h_i(S^j) \right| = \left| \mathbb{E}_{\mathbf{z}'_i \sim \mathcal{D}} \mathbb{E}_{S_G \sim \mathcal{D}_G^{m_G}(S^i)} \left[ \mathbb{E}_{\mathbf{z} \sim \mathcal{D}} \ell(\mathcal{A}(S^i \cup S_G), \mathbf{z}) - \ell(\mathcal{A}(S^i \cup S_G), \mathbf{z}_i) \right] \right.$$
$$\left. - \mathbb{E}_{\mathbf{z}'_i \sim \mathcal{D}} \mathbb{E}_{S_G \sim \mathcal{D}_G^{m_G}((S^j)^i)} \left[ \mathbb{E}_{\mathbf{z} \sim \mathcal{D}} \ell(\mathcal{A}((S^j)^i \cup S_G), \mathbf{z}) - \ell(\mathcal{A}((S^j)^i \cup S_G), \mathbf{z}_i) \right] \right|$$
$$\leq \left| \mathbb{E}_{\mathbf{z}'_i \sim \mathcal{D}} \mathbb{E}_{S_G \sim \mathcal{D}_G^{m_G}(S^i)} \left[ \mathbb{E}_{\mathbf{z} \sim \mathcal{D}} \ell(\mathcal{A}(S^i \cup S_G), \mathbf{z}) - \ell(\mathcal{A}(S^i \cup S_G), \mathbf{z}_i) \right] \right.$$
$$\left. - \mathbb{E}_{\mathbf{z}'_i \sim \mathcal{D}} \mathbb{E}_{S_G \sim \mathcal{D}_G^{m_G}(S^i)} \left[ \mathbb{E}_{\mathbf{z} \sim \mathcal{D}} \ell(\mathcal{A}((S^j)^i \cup S_G), \mathbf{z}) - \ell(\mathcal{A}((S^j)^i \cup S_G), \mathbf{z}_i) \right] \right|$$
$$\tag{8}$$
$$+ \left| \mathbb{E}_{\mathbf{z}'_i \sim \mathcal{D}} \mathbb{E}_{S_G \sim \mathcal{D}_G^{m_G}(S^i)} \left[ \mathbb{E}_{\mathbf{z} \sim \mathcal{D}} \ell(\mathcal{A}((S^j)^i \cup S_G), \mathbf{z}) - \ell(\mathcal{A}((S^j)^i \cup S_G), \mathbf{z}_i) \right] \right.$$
$$\left. - \mathbb{E}_{\mathbf{z}'_i \sim \mathcal{D}} \mathbb{E}_{S_G \sim \mathcal{D}_G^{m_G}((S^j)^i)} \left[ \mathbb{E}_{\mathbf{z} \sim \mathcal{D}} \ell(\mathcal{A}((S^j)^i \cup S_G), \mathbf{z}) - \ell(\mathcal{A}((S^j)^i \cup S_G), \mathbf{z}_i) \right] \right|.$$
$$\tag{9}$$

We bound (8) and (9) respectively. The first can be bounded by using the property of uniform stability.

$$\left| \mathbb{E}_{\mathbf{z}'_i \sim \mathcal{D}} \mathbb{E}_{S_G \sim \mathcal{D}_G^{m_G}(S^i)} \left[ \mathbb{E}_{\mathbf{z} \sim \mathcal{D}} \ell(\mathcal{A}(S^i \cup S_G), \mathbf{z}) - \ell(\mathcal{A}(S^i \cup S_G), \mathbf{z}_i) \right] \right.$$
$$\left. - \mathbb{E}_{\mathbf{z}'_i \sim \mathcal{D}} \mathbb{E}_{S_G \sim \mathcal{D}_G^{m_G}(S^i)} \left[ \mathbb{E}_{\mathbf{z} \sim \mathcal{D}} \ell(\mathcal{A}((S^j)^i \cup S_G), \mathbf{z}) - \ell(\mathcal{A}((S^j)^i \cup S_G), \mathbf{z}_i) \right] \right|$$
$$= \left| \mathbb{E}_{\mathbf{z}'_i \sim \mathcal{D}} \mathbb{E}_{S_G \sim \mathcal{D}_G^{m_G}(S^i)} \left[ \mathbb{E}_{\mathbf{z} \sim \mathcal{D}} \ell(\mathcal{A}(S^i \cup S_G), \mathbf{z}) - \ell(\mathcal{A}(S^i \cup S_G), \mathbf{z}_i) \right. \right.$$
$$\left. \left. - \mathbb{E}_{\mathbf{z} \sim \mathcal{D}} \ell(\mathcal{A}((S^j)^i \cup S_G), \mathbf{z}) + \ell(\mathcal{A}((S^j)^i \cup S_G), \mathbf{z}_i) \right] \right|$$
$$\leq \left| \mathbb{E}_{\mathbf{z}'_i \sim \mathcal{D}} \mathbb{E}_{S_G \sim \mathcal{D}_G^{m_G}(S^i)} \mathbb{E}_{\mathbf{z} \sim \mathcal{D}} \left[ \ell(\mathcal{A}(S^i \cup S_G), \mathbf{z}) - \ell(\mathcal{A}((S^j)^i \cup S_G), \mathbf{z}) \right] \right|$$
$$+ \left| \mathbb{E}_{\mathbf{z}'_i \sim \mathcal{D}} \mathbb{E}_{S_G \sim \mathcal{D}_G^{m_G}(S^i)} \left[ \ell(\mathcal{A}(S^i \cup S_G), \mathbf{z}_i) - \ell(\mathcal{A}((S^j)^i \cup S_G), \mathbf{z}_i) \right] \right|$$
$$\leq \mathbb{E}_{\mathbf{z}'_i \sim \mathcal{D}} \mathbb{E}_{S_G \sim \mathcal{D}_G^{m_G}(S^i)} \mathbb{E}_{\mathbf{z} \sim \mathcal{D}} \left| \ell(\mathcal{A}(S^i \cup S_G), \mathbf{z}) - \ell(\mathcal{A}((S^j)^i \cup S_G), \mathbf{z}) \right|$$
$$+ \mathbb{E}_{\mathbf{z}'_i \sim \mathcal{D}} \mathbb{E}_{S_G \sim \mathcal{D}_G^{m_G}(S^i)} \left| \ell(\mathcal{A}(S^i \cup S_G), \mathbf{z}_i) - \ell(\mathcal{A}((S^j)^i \cup S_G), \mathbf{z}_i) \right|$$
$$\leq \beta_{m_T} + \beta_{m_T} = 2\beta_{m_T}.$$

We denote $\ell(\mathcal{A}((S^j)^i \cup S_G), \mathbf{z}) - \ell(\mathcal{A}((S^j)^i \cup S_G), \mathbf{z}_i)$ by $B$ for convenience, then we have

$$\left| \mathbb{E}_{\mathbf{z}'_i \sim \mathcal{D}} \mathbb{E}_{S_G \sim \mathcal{D}_G^{m_G}(S^i)} \left[ \mathbb{E}_{\mathbf{z} \sim \mathcal{D}} \ell(\mathcal{A}((S^j)^i \cup S_G), \mathbf{z}) - \ell(\mathcal{A}((S^j)^i \cup S_G), \mathbf{z}_i) \right] \right.$$

$$- \mathbb{E}_{\mathbf{z}'_i \sim \mathcal{D}} \mathbb{E}_{S_G \sim \mathcal{D}_G^{m_G}((S^j)^i)} \left[ \mathbb{E}_{\mathbf{z} \sim \mathcal{D}} \ell(\mathcal{A}((S^j)^i \cup S_G), \mathbf{z}) - \ell(\mathcal{A}((S^j)^i \cup S_G), \mathbf{z}_i) \right] \Bigg|$$

$$= \Bigg| \mathbb{E}_{\mathbf{z}'_i \sim \mathcal{D}} \mathbb{E}_{\mathbf{z} \sim \mathcal{D}} \mathbb{E}_{S_G \sim \mathcal{D}_G^{m_G}(S^i)} \left[ \ell(\mathcal{A}((S^j)^i \cup S_G), \mathbf{z}) - \ell(\mathcal{A}((S^j)^i \cup S_G), \mathbf{z}_i) \right]$$

$$- \mathbb{E}_{\mathbf{z}'_i \sim \mathcal{D}} \mathbb{E}_{\mathbf{z} \sim \mathcal{D}} \mathbb{E}_{S_G \sim \mathcal{D}_G^{m_G}((S^j)^i)} \left[ \ell(\mathcal{A}((S^j)^i \cup S_G), \mathbf{z}) - \ell(\mathcal{A}((S^j)^i \cup S_G), \mathbf{z}_i) \right] \Bigg|$$

$$= \Bigg| \mathbb{E}_{\mathbf{z}'_i \sim \mathcal{D}} \mathbb{E}_{\mathbf{z} \sim \mathcal{D}} \mathbb{E}_{S_G \sim \mathcal{D}_G^{m_G}(S^i)} [B] - \mathbb{E}_{\mathbf{z}'_i \sim \mathcal{D}} \mathbb{E}_{\mathbf{z} \sim \mathcal{D}} \mathbb{E}_{S_G \sim \mathcal{D}_G^{m_G}((S^j)^i)} [B] \Bigg|$$

$$= \Bigg| \mathbb{E}_{\mathbf{z}'_i \sim \mathcal{D}} \mathbb{E}_{\mathbf{z} \sim \mathcal{D}} \left[ \mathbb{E}_{S_G \sim \mathcal{D}_G^{m_G}(S^i)} [B] - \mathbb{E}_{S_G \sim \mathcal{D}_G^{m_G}((S^j)^i)} [B] \right] \Bigg|$$

$$\leq \mathbb{E}_{\mathbf{z}'_i \sim \mathcal{D}} \mathbb{E}_{\mathbf{z} \sim \mathcal{D}} \Bigg| \mathbb{E}_{S_G \sim \mathcal{D}_G^{m_G}(S^i)} [B] - \mathbb{E}_{S_G \sim \mathcal{D}_G^{m_G}((S^j)^i)} [B] \Bigg|$$

$$= \mathbb{E}_{\mathbf{z}'_i \sim \mathcal{D}} \mathbb{E}_{\mathbf{z} \sim \mathcal{D}} \Bigg| \int_{S_G} \left( \mathbb{P}(S_G | S^i) - \mathbb{P}(S_G | (S^j)^i) \right) B \, dS_G \Bigg|$$

$$\leq \mathbb{E}_{\mathbf{z}'_i \sim \mathcal{D}} \mathbb{E}_{\mathbf{z} \sim \mathcal{D}} \left[ \int_{S_G} \Bigg| \left( \mathbb{P}(S_G | S^i) - \mathbb{P}(S_G | (S^j)^i) \right) B \Bigg| \, dS_G \right]$$

$$\leq M \mathbb{E}_{\mathbf{z}'_i \sim \mathcal{D}} \mathbb{E}_{\mathbf{z} \sim \mathcal{D}} \left[ \int_{S_G} \Bigg| \mathbb{P}(S_G | S^i) - \mathbb{P}(S_G | (S^j)^i) \Bigg| \, dS_G \right]$$

$$\leq 2M \sup_i \mathcal{D}_{\mathrm{TV}} \left( \mathcal{D}_G^{m_G}(S^i), \mathcal{D}_G^{m_G}(S) \right) = 2M \mathfrak{T}(m_S, m_G).$$

Therefore, $h_i$ has a bounded difference $2\beta_{m_T} + 2M\mathfrak{T}(m_S, m_G)$ with respect to all variables except the $i$-th variable. By Lemma B.3, we have

$$\left\| \sum_{i=1}^{m_S} h_i(S) \right\|_p \leq 12\sqrt{2} p m_S \left( 2\beta_{m_T} + 2M\mathfrak{T}(m_S, m_G) \right) \log m_S + 4M\sqrt{p m_S} \tag{10}$$

$$\lesssim p m_S \left( \beta_{m_T} + M\mathfrak{T}(m_S, m_G) \right) \log m_S + M\sqrt{p m_S}. \tag{11}$$

We note the gap between $\Delta_2$ and $\left\| \sum_{i=1}^{m_S} h_i(S) \right\|_p$ is small, then we can bound $\Delta_2$ by (10) as follows.

$$\Delta_2 = \left\| \mathbb{E}_{S_G \sim \mathcal{D}_G^{m_G}(S)} \Phi_1 \right\|_p$$

$$= \left\| \mathbb{E}_{S_G \sim \mathcal{D}_G^{m_G}(S)} \left[ m_S \mathcal{R}_{\mathcal{D}}(\mathcal{A}(\widetilde{S})) - \sum_{i=1}^{m_S} \ell(\mathcal{A}(\widetilde{S}), \mathbf{z}_i) \right] \right\|_p$$

$$= \left\| \sum_{i=1}^{m_S} \mathbb{E}_{S_G \sim \mathcal{D}_G^{m_G}(S)} \left[ m_S \mathcal{R}_{\mathcal{D}}(\mathcal{A}(\widetilde{S})) - \ell(\mathcal{A}(\widetilde{S}), \mathbf{z}_i) \right] \right\|_p$$

$$\leq \left\| \sum_{i=1}^{m_S} \left( \mathbb{E}_{\mathbf{z}'_i \sim \mathcal{D}} \mathbb{E}_{S_G \sim \mathcal{D}_G^{m_G}(S^i)} \left[ \mathbb{E}_{\mathbf{z} \sim \mathcal{D}} \ell(\mathcal{A}(S^i \cup S_G), \mathbf{z}) - \ell(\mathcal{A}(S^i \cup S_G), \mathbf{z}_i) \right] \right) \right\|_p \tag{12}$$

$$+ \left\| 2m_S \beta_{m_T} + 2m_S M \sup_i \mathcal{D}_{\mathrm{TV}} \left( \mathcal{D}_G^{m_G}(S), \mathcal{D}_G^{m_G}(S^i) \right) \right\|_p$$

$$= \left\| \sum_{i=1}^{m_S} h_i(S) \right\|_p + \left\| 2m_S \beta_{m_T} + 2m_S M\mathfrak{T}(m_S, m_G) \right\|_p$$

$$\lesssim pm_S\left(\beta_{m_T} + M\mathcal{T}(m_S, m_G)\right)\log m_S + M\sqrt{pm_S}$$
$$+ m_S\beta_{m_T} + m_S M\mathcal{T}(m_S, m_G)$$
$$\lesssim pm_S\left(\beta_{m_T} + M\mathcal{T}(m_S, m_G)\right)\log m_S + M\sqrt{pm_S}. \tag{13}$$

Combine (7) and (13), we have

$$\left\|\Phi_1(S, S_G)\right\|_p \lesssim \sqrt{m_G p}\, m_S\beta_{m_T} + pm_S\left(\beta_{m_T} + M\mathcal{T}(m_S, m_G)\right)\log m_S + M\sqrt{pm_S}$$
$$= \sqrt{p}\left(M\sqrt{m_S} + \sqrt{m_G}m_S\beta_{m_T}\right) + pm_S\left(\beta_{m_T} + M\mathcal{T}(m_S, m_G)\right)\log m_S \tag{14}$$

In addition, by (14) and (6), we have

$$\left\|m_T\Phi(S, S_G)\right\|_p \lesssim \sqrt{p}\left(M\sqrt{m_S} + M\sqrt{m_G} + \sqrt{m_G}m_S\beta_{m_T}\right)$$
$$+ p\left(m_S\beta_{m_T}\log m_S + m_G\beta_{m_T}\log m_G + m_S\log m_S M\mathcal{T}(m_S, m_G)\right).$$

By Lemma B.1, we can bound the generalization error w.r.t. mixed distribution $\left|\Phi(S, S_G)\right| = \left|\mathcal{R}_{\widetilde{\mathcal{D}}(S)}(\mathcal{A}(\widetilde{S})) - \widehat{\mathcal{R}}_{\widetilde{S}}(\mathcal{A}(\widetilde{S}))\right|$ as follows.

$$\left|\mathcal{R}_{\widetilde{\mathcal{D}}(S)}(\mathcal{A}(\widetilde{S})) - \widehat{\mathcal{R}}_{\widetilde{S}}(\mathcal{A}(\widetilde{S}))\right|$$
$$\lesssim \frac{M(\sqrt{m_S} + \sqrt{m_G}) + m_S\sqrt{m_G}\beta_{m_T}}{m_T}\sqrt{\log\left(\frac{1}{\delta}\right)}$$
$$+ \frac{\beta_{m_T}\left(m_S\log m_S + m_G\log m_G\right) + m_S\log m_S M\mathcal{T}(m_S, m_G)}{m_T}\log\left(\frac{1}{\delta}\right).$$

Finally, we conclude that

$$\left|\mathcal{R}_{\mathcal{D}}(\mathcal{A}(\widetilde{S})) - \widehat{\mathcal{R}}_{\widetilde{S}}(\mathcal{A}(\widetilde{S}))\right|$$
$$\lesssim \frac{m_G}{m_T}M\mathcal{D}_{\mathrm{TV}}\left(\mathcal{D}, \mathcal{D}_G(S)\right) + \frac{M(\sqrt{m_S} + \sqrt{m_G}) + m_S\sqrt{m_G}\beta_{m_T}}{m_T}\sqrt{\log\left(\frac{1}{\delta}\right)}$$
$$+ \frac{\beta_{m_T}\left(m_S\log m_S + m_G\log m_G\right) + m_S\log m_S M\mathcal{T}(m_S, m_G)}{m_T}\log\left(\frac{1}{\delta}\right)$$
$$\lesssim \frac{m_G}{m_T}M\mathcal{D}_{\mathrm{TV}}\left(\mathcal{D}, \mathcal{D}_G(S)\right) + \frac{M(\sqrt{m_S} + \sqrt{m_G}) + m_S\sqrt{m_G}\beta_{m_T}}{m_T}\sqrt{\log\left(\frac{1}{\delta}\right)}$$
$$+ \frac{\beta_{m_T}\left(m_S\log m_S + m_G\log m_G\right) + m_S\log m_S M\mathcal{T}(m_S, m_G)}{m_T}\log\left(\frac{1}{\delta}\right),$$

which completes the proof.

$\square$

## B.2 Proof of Theorem 3.2

We need to bound terms $M$, $\beta_{m_T}$, $\mathcal{D}_{\mathrm{TV}}\left(\mathcal{D}, \mathcal{D}_G(S)\right)$ and $\mathcal{T}(m_S, m_G)$ in Theorem 3.1. For $M$ (Lemma B.5) and $\beta_{m_T}$ (Lemma B.6), we mainly use the boundedness of the multivariate Gaussian variable with high probability (Lemma B.4). In addition, we bound $\mathcal{D}_{\mathrm{TV}}\left(\mathcal{D}, \mathcal{D}_G(S)\right)$ (Lemma B.7) by discussing the distance between the estimated parameters and the true parameters of bGMM. Besides, the concentration property of $\mathcal{T}(m_S, m_G)$ (Lemma B.9) can be induced by the preceding discussion.

**Lemma B.4** ("Boundedness" of multivariate Gaussian distribution). *Let $\mathbf{X} = (X_1, \ldots, X_d)$ be a d-dimension isotropic Gaussian random variable, which satisfies $\|\boldsymbol{\mu}\|_2 = 1$ and $\sigma_i^2 = \sigma^2$ for any $i \in \{1, \ldots, d\}$. For any $\delta \in (0, 1)$, with probability at least $1 - \delta$, it holds that*

$$\|\mathbf{X}\|_2 \lesssim \sigma \sqrt{d + \log(\frac{1}{\delta})}.$$

*Proof.* The proof idea is to bound the distance between $\|\mathbf{X}\|_2^2$ and its expectation with high probability. Let $\mathbf{Z}$ be the standard $d$-dimension isotropic Gaussian random variable, we have

$$\mathbb{P}\left(\left|\frac{\|\mathbf{X}\|_2^2}{d} - \sigma^2 - \frac{1}{d}\right| \geq \epsilon\right)$$

$$= \mathbb{P}\left(\left|\frac{1}{d}\sum_{i=1}^{d}\left(X_i^2 - \sigma^2 - \mu_i^2\right)\right| \geq \epsilon\right)$$

$$= \mathbb{P}\left(\left|\frac{1}{d}\sum_{i=1}^{d}\left((\sigma Z_i + \mu_i)^2 - \sigma^2 - \mu_i^2\right)\right| \geq \epsilon\right)$$

$$= \mathbb{P}\left(\left|\frac{1}{d}\sum_{i=1}^{d}\left(\sigma^2(Z_i^2 - 1) + 2\sigma\mu_i Z_i\right)\right| \geq \epsilon\right)$$

$$\leq \mathbb{P}\left(\left|\frac{1}{d}\sum_{i=1}^{d}\left(\sigma^2(Z_i^2 - 1)\right)\right| + \left|\frac{1}{d}\sum_{i=1}^{d}(2\sigma\mu_i Z_i)\right| \geq \epsilon\right)$$

$$\leq \mathbb{P}\left(\left|\frac{1}{d}\sum_{i=1}^{d}\left(\sigma^2(Z_i^2 - 1)\right)\right| \geq \frac{\epsilon}{2} \cup \left|\frac{1}{d}\sum_{i=1}^{d}(2\sigma\mu_i Z_i)\right| \geq \frac{\epsilon}{2}\right)$$

$$\leq \mathbb{P}\left(\left|\frac{1}{d}\sum_{i=1}^{d}\left(\sigma^2(Z_i^2 - 1)\right)\right| \geq \frac{\epsilon}{2}\right) + \mathbb{P}\left(\left|\frac{1}{d}\sum_{i=1}^{d}(2\sigma\mu_i Z_i)\right| \geq \frac{\epsilon}{2}\right)$$

$$= \mathbb{P}\left(\left|\frac{1}{d}\sum_{i=1}^{d}\left(Z_i^2 - 1\right)\right| \geq \frac{\epsilon}{2\sigma^2}\right) + \mathbb{P}\left(\left|\frac{1}{d}\sum_{i=1}^{d}\mu_i Z_i\right| \geq \frac{\epsilon}{4\sigma}\right).$$

We bound each of the two terms respectively. For the first term, we note that $Z_i^2$ obeys $\chi^2(1)$ distribution and is a sub-exponential random variable, so it can be bounded by using Bernstein's inequality (e.g., Proposition 2.9, [78]). By Example 2.8 in [78], for any $\lambda \in (0, 1/4)$, we have

$$\mathbb{E}\left[\exp\left(\lambda(Z_i^2 - 1)\right)\right] = \frac{\exp(-\lambda)}{\sqrt{1 - 2\lambda}} \leq \exp(2\lambda^2).$$

In addition, through Bernstein's inequality, we have

$$\mathbb{P}\left(\left|\frac{1}{d}\sum_{i=1}^{d}\left(Z_i^2 - 1\right)\right| \geq \frac{\epsilon}{2\sigma^2}\right) \leq \begin{cases} 2\exp(-\frac{d\epsilon^2}{32\sigma^4}) & \text{if } 0 \leq \epsilon \leq 2\sigma^2, \\ 2\exp(-\frac{d\epsilon}{32\sigma^2}) & \text{if } \epsilon > 2\sigma^2. \end{cases}$$

For the second term, we bound it directly by using Hoeffding's inequality (e.g., Proposition 2.5, [78]).

$$\mathbb{P}\left(\left|\frac{1}{d}\sum_{i=1}^{d}\mu_i Z_i\right| \geq \frac{\epsilon}{4\sigma}\right) \leq 2\exp(-\frac{d\epsilon^2}{32\sigma^4 \sum_{i=1}^{d}\mu_i^2}) = 2\exp(-\frac{d\epsilon^2}{32\sigma^4}).$$

Therefore, for any $\epsilon \leq 2\sigma^2$, we have

$$\mathbb{P}\left(\left|\|\mathbf{X}\|_2^2 - \sigma^2 d - 1\right| \geq d\epsilon\right) = \mathbb{P}\left(\left|\frac{\|\mathbf{X}\|_2^2}{d} - \sigma^2 - \frac{1}{d}\right| \geq \epsilon\right) \leq 4\exp(-\frac{d\epsilon^2}{32\sigma^4}).$$

Let $4\exp(-\frac{d\epsilon^2}{32\sigma^4}) = \delta$, then with probability at least $1 - \delta$, it holds that

$$\|\mathbf{X}\|_2^2 \leq \sigma^2 d + 1 + d\sigma^2\sqrt{\frac{32}{d}\log(\frac{4}{\delta})} \lesssim \sigma^2\left(d + \sqrt{d\log(\frac{1}{\delta})}\right)$$

which means that

$$\|\mathbf{X}\|_2 \lesssim \sigma\sqrt{d + \sqrt{d\log(\frac{1}{\delta})}} \leq \sigma\sqrt{d + \frac{1}{2}d + \frac{1}{2}\log(\frac{1}{\delta})} \lesssim \sigma\sqrt{d + \log(\frac{1}{\delta})}.$$

Similarly, for any $\epsilon > 2\sigma^2$, we have

$$\mathbb{P}\left(\left|\frac{\|\mathbf{X}\|_2^2}{d} - \sigma^2 - \frac{1}{d}\right| \geq \epsilon\right) \leq 2\exp(-\frac{d\epsilon}{32\sigma^2}) + 2\exp(-\frac{d\epsilon^2}{32\sigma^4})$$

$$\leq 2\exp(-\frac{d\epsilon}{32\sigma^2}) + 2\exp(-\frac{d\epsilon}{16\sigma^2})$$

$$\leq 4\exp(-\frac{d\epsilon}{32\sigma^2}).$$

Let $4\exp(-\frac{d\epsilon}{32\sigma^2}) = \delta$, then with probability at least $1 - \delta$, it holds that

$$\|\mathbf{X}\|_2^2 \leq \sigma^2 d + 1 + d\sigma^2\frac{32}{d}\log(\frac{4}{\delta}) \lesssim \sigma^2\left(d + \log(\frac{1}{\delta})\right),$$

which also implies

$$\|\mathbf{X}\|_2 \lesssim \sigma\sqrt{d + \log(\frac{4}{\delta})} \leq \sigma\sqrt{d + \log(\frac{1}{\delta})}.$$

The proof is completed.

$\square$

Based on the "boundedness" of multivariate Gaussian distribution, we can bound $M$, $\beta_m$, $\mathcal{D}_{\text{TV}}(\mathcal{D}_G(S), \mathcal{D}_G)$ and $\mathcal{T}(m_S, m_G)$, respectively. They are listed as the following.

**Lemma B.5** (Concentration bound for $M$). *For any $\delta \in (0, 1)$, with probability at least $1 - \delta$, it holds that*

$$\left|\ell(\mathcal{A}(S), \mathbf{z})\right| \lesssim d + \log(\frac{m}{\delta}).$$

*Proof.* Given a set $S = \{(\mathbf{x}_1, y_1), \ldots, (\mathbf{x}_m, y_m)\}$ and $\mathbf{z}$ sampled from binary mixture Gaussian distribution, by Lemma B.4, we know that for any $\delta \in (0, 1)$, with probability at least $1 - \delta$,

$$\max_i \|\mathbf{x}_i\|_2 \lesssim \sigma\sqrt{d + \log(\frac{m + 1}{\delta})}.$$

Under this condition, we have

$$\left|\ell(\mathcal{A}(S), \mathbf{z})\right|$$

$$= \left| \frac{1}{2\sigma^2} (\mathbf{x} - y\boldsymbol{\theta})^\top (\mathbf{x} - y\boldsymbol{\theta}) \right|$$

$$= \frac{1}{2\sigma^2} \left| \mathbf{x}^\top \mathbf{x} - 2y\mathbf{x}^\top \theta + \theta^\top \theta \right|$$

$$\leq \frac{1}{2\sigma^2} \left( \left| \mathbf{x}^\top \mathbf{x} \right| + 2\left| \mathbf{x}^\top \theta \right| + \left| \theta^\top \theta \right| \right)$$

$$\leq \frac{1}{2\sigma^2} \left( \|\mathbf{x}\|_2^2 + 2\|\mathbf{x}\|_2\|\theta\|_2 + \|\theta\|_2^2 \right)$$

$$= \frac{1}{2\sigma^2} \left( \|\mathbf{x}\|_2^2 + 2\|\mathbf{x}\|_2\|\frac{1}{m}\sum_{i=1}^{m} y_i\mathbf{x}_i\|_2 + \|\frac{1}{m}\sum_{i=1}^{m} y_i\mathbf{x}_i\|_2^2 \right)$$

$$\leq \frac{1}{2\sigma^2} \left( \|\mathbf{x}\|_2^2 + 2\frac{1}{m}\sum_{i=1}^{m} \|\mathbf{x}\|_2\|\mathbf{x}_i\|_2 + \left(\frac{1}{m}\sum_{i=1}^{m} \|\mathbf{x}_i\|_2\right)^2 \right)$$

$$\lesssim \frac{1}{2\sigma^2} \left( \sigma^2\left(d + \log(\frac{m+1}{\delta})\right) + \frac{2}{m}\sum_{i=1}^{m} \sigma^2\left(d + \log(\frac{m+1}{\delta})\right) + \left(\frac{1}{m}\sum_{i=1}^{m} \sigma\sqrt{d + \log(\frac{m+1}{\delta})}\right)^2 \right)$$

$$= \frac{1}{2\sigma^2}4\sigma^2\left(d + \log(\frac{m+1}{\delta})\right) = 2\left(d + \log(\frac{m+1}{\delta})\right)$$

$$\lesssim d + \log(\frac{m}{\delta}).$$

$\square$

**Lemma B.6** (Concentration bound for $\beta_m$). *For any $\delta \in (0,1)$, with probability at least $1 - \delta$, it holds that*

$$\left| \ell(\mathcal{A}(S), \mathbf{z}) - \ell(\mathcal{A}(S^i), \mathbf{z}) \right| \lesssim \frac{1}{m}\left(d + \log(\frac{m}{\delta})\right).$$

*Proof.* Given $m+2$ samples $S$, $\mathbf{z}$ and $\mathbf{z}'_i$ randomly sampled from binary mixture Gaussian distribution, for any $\delta \in (0,1)$, with probability at least $1 - \delta$, we have

$$\left| \ell(\mathcal{A}(S), \mathbf{z}) - \ell(\mathcal{A}(S^i), \mathbf{z}) \right|$$

$$= \left| \frac{1}{2\sigma^2}(\mathbf{x} - y\boldsymbol{\theta})^\top(\mathbf{x} - y\boldsymbol{\theta}) - \frac{1}{2\sigma^2}(\mathbf{x} - y\boldsymbol{\theta}')^\top(\mathbf{x} - y\boldsymbol{\theta}') \right|$$

$$= \frac{1}{2\sigma^2}\left| 2y\left(\mathbf{x}^\top \theta' - \mathbf{x}^\top \theta\right) + \theta^\top \theta - \theta'^\top \theta' \right|$$

$$= \frac{1}{2\sigma^2}\left| 2y\left(\mathbf{x}^\top \theta' - \mathbf{x}^\top \theta\right) + (\theta + \theta')^\top(\theta - \theta') \right|$$

$$\leq \frac{1}{2\sigma^2}\left( 2\left|\left(\mathbf{x}^\top(\theta' - \theta)\right)\right| + \left|(\theta + \theta')^\top(\theta - \theta')\right| \right)$$

$$\leq \frac{1}{2\sigma^2}\left( 2\|\mathbf{x}\|_2\|\theta' - \theta\|_2 + \|\theta + \theta'\|_2\|\theta - \theta'\|_2 \right)$$

$$= \frac{1}{2\sigma^2}\left( 2\|\mathbf{x}\|_2 + \|\theta + \theta'\|_2 \right)\|\theta' - \theta\|_2$$

$$= \frac{1}{2\sigma^2}\left( 2\|\mathbf{x}\|_2 + \|\theta + \theta'\|_2 \right)\|\frac{1}{m}(y_i\mathbf{x}_i - y'_i\mathbf{x}'_i)\|_2$$

$$\leq \frac{1}{2m\sigma^2}\left( 2\|\mathbf{x}\|_2 + \|\theta\|_2 + \|\theta'\|_2 \right)\left( \|\mathbf{x}_i\|_2 + \|\mathbf{x}'_i\|_2 \right)$$

$$\lesssim \frac{8}{2m\sigma^2}\sigma^2\left(d + \log(\frac{m+2}{\delta})\right)$$

$$\lesssim \frac{4}{m}\left(d + \log(\frac{m}{\delta})\right) \lesssim \frac{1}{m}\left(d + \log(\frac{m}{\delta})\right).$$

$\square$

**Lemma B.7** (Concentration bound for $\mathcal{D}_{\mathrm{TV}}(\mathcal{D}, \mathcal{D}_G(S))$)**.** *With high probability at least $1 - \delta$, it holds that*

$$\mathcal{D}_{\mathrm{TV}}(\mathcal{D}, \mathcal{D}_G(S)) \lesssim \min\left(1, \sqrt{\frac{d}{m}\log\left(\frac{d}{\delta}\right)}\right).$$

The idea of the proof of Lemma B.7 built upon the estimation for Gaussian distribution. As the sample size increases, parameters can be estimated more accurately, which leads to a smaller distance between the estimated and true Gaussian distributions. The concentration bound of the estimated parameters can be inscribed by the following lemma.

**Lemma B.8.** *Let $m = O\left(\frac{1}{\epsilon^2}\log\left(\frac{d}{\delta}\right)\right)$, then with high probability at least $1 - \delta$, for any $i \in \{1, \ldots, d\}$, it holds that*

$$\left|\frac{\widehat{\sigma_i^2}}{\sigma^2} - 1\right| \leq \epsilon, \quad \frac{|\widehat{\mu_{yi}} - \mu_{yi}|}{\sigma} \leq \epsilon.$$

*Proof.* Let $\epsilon \leq 1/4$, and $m_y$ be the number of samples from category $y$. By Hoeffding's inequality (Proposition 2.5, [78]), we have

$$\mathbb{P}\left(\left|m_y - \frac{m}{2}\right| \geq m\epsilon\right) \leq 2\exp(-\frac{m^2\epsilon^2}{2m(1/2)^2}) = 2\exp(-2m\epsilon^2) = \delta_1,$$

which means $m_y \geq m/2 - \epsilon m \geq m/4$, and $m_y \leq m/2 + \epsilon m \leq 3m/4$. We can bound $\widehat{\sigma}_i^2$ and $\widehat{\mu}_{yi}$ based on the concentration property of $m_y$. In terms of $\widehat{\mu}_{yi}$, give a fixed $m_y$, we can write

$$\mathbb{P}\left(\frac{|\widehat{\mu_{yi}} - \mu_{yi}|}{\sigma} \geq \epsilon \mid m_y\right) = \mathbb{P}\left(\frac{1}{\sigma}\left|\frac{\sum_{y_i=y} x_i}{m_y} - \mu_{yi}\right| \geq \epsilon\right)$$

$$= \mathbb{P}\left(\left|\sum_{y_i=y} x_i - m_y\mu_{yi}\right| \geq \sigma m_y \epsilon\right)$$

$$\leq \exp\left(-\frac{\sigma^2 m_y^2 \epsilon^2}{2m_y\sigma^2}\right) = \exp\left(-\frac{m_y\epsilon^2}{2}\right).$$

Furthermore, by the law of total probability, we have

$$\mathbb{P}\left(\frac{|\widehat{\mu_{yi}} - \mu_{yi}|}{\sigma} \geq \epsilon\right)$$

$$= \mathbb{P}\left(\frac{|\widehat{\mu_{yi}} - \mu_{yi}|}{\sigma} \geq \epsilon \mid m_y \geq m/2 - \epsilon m\right)\mathbb{P}(m_y \geq m/2 - \epsilon m)$$

$$+ \mathbb{P}\left(\frac{|\widehat{\mu_{yi}} - \mu_{yi}|}{\sigma} \geq \epsilon \mid m_y \leq m/2 - \epsilon m\right)\mathbb{P}(m_y \leq m/2 - \epsilon m)$$

$$\leq \exp\left(-\frac{(m/4)\epsilon^2}{2}\right) + \delta_1 = \exp\left(-\frac{m\epsilon^2}{8}\right) + \delta_1 = \delta_2.$$

For the estimation of $\widehat{\sigma}_i^2$, we can obtain its concentration bound in a similar way.

$$\mathbb{P}\left(\left|\frac{\widehat{\sigma_i^2}}{\sigma^2} - 1\right| \geq \epsilon \mid m_y\right) = \mathbb{P}\left(\left|\sum_y \frac{m_y}{m\sigma^2} \frac{\sum_{y_i=y}(x_i - \widehat{\mu}_{yi})^2}{m_y - 1} - 1\right| \geq \epsilon\right)$$

$$= \mathbb{P}\left(\left|\sum_y \frac{m_y}{m}\left(\frac{\sum_{y_i=y}(x_i - \widehat{\mu}_{yi})^2}{(m_y - 1)\sigma^2} - 1\right)\right| \geq \epsilon\right)$$

$$\leq \mathbb{P}\left(\sum_y \left|\frac{m_y}{m}\left(\frac{\sum_{y_i=y}(x_i - \widehat{\mu}_{yi})^2}{(m_y - 1)\sigma^2} - 1\right)\right| \geq \epsilon\right)$$

$$\leq \mathbb{P}\left(\cup_{y=\{-1,1\}}\left|\frac{m_y}{m}\left(\frac{\sum_{y_i=y}(x_i - \widehat{\mu}_{yi})^2}{(m_y - 1)\sigma^2} - 1\right)\right| \geq \epsilon/2\right)$$

$$\leq \sum_y \mathbb{P}\left(\left|\frac{m_y}{m}\left(\frac{\sum_{y_i=y}(x_i - \widehat{\mu}_{yi})^2}{(m_y - 1)\sigma^2} - 1\right)\right| \geq \epsilon/2\right)$$

$$\leq \sum_y \mathbb{P}\left(\left|\frac{m_y}{m}\left(\frac{\sum_{y_i=y}(x_i - \widehat{\mu}_{yi})^2}{\sigma^2} - (m_y - 1)\right)\right| \geq (m_y - 1)\epsilon/2\right)$$

$$= \sum_y \mathbb{P}\left(\left|\frac{\sum_{y_i=y}(x_i - \widehat{\mu}_{yi})^2}{\sigma^2} - (m_y - 1)\right| \geq \frac{(m_y - 1)m}{2m_y}\epsilon\right)$$

$$= \sum_y \mathbb{P}\left(\left|\chi^2(m_y - 1) - (m_y - 1)\right| \geq \frac{(m_y - 1)m}{2m_y}\epsilon\right)$$

$$= \sum_y \mathbb{P}\left(\left|\frac{1}{m_y - 1}\sum_{i=1}^{m_y-1}\chi^2(1) - 1\right| \geq \frac{m}{2m_y}\epsilon\right)$$

$$\leq \sum_y 2\exp\left(-\frac{m_y - 1}{8}(\frac{m}{2m_y}\epsilon)^2\right) \quad \text{(Bernstein's inequality)}$$

$$= \sum_y 2\exp\left(-\frac{(m_y - 1)m^2\epsilon^2}{32m_y^2}\right)$$

Without loss of generality, we assume that $m \geq 8$, then by the law of total probability, it holds that

$$\mathbb{P}\left(\left|\frac{\widehat{\sigma_i^2}}{\sigma^2} - 1\right| \geq \epsilon\right) = \mathbb{P}\left(\left|\frac{\widehat{\sigma_i^2}}{\sigma^2} - 1\right| \geq \epsilon \mid \left|m_y - m/2\right| \leq \epsilon m\right)\mathbb{P}\left(\left|m_y - m/2\right| \leq \epsilon m\right)$$

$$+ \mathbb{P}\left(\left|\frac{\widehat{\sigma_i^2}}{\sigma^2} - 1\right| \geq \epsilon \mid \left|m_y - m/2\right| \geq \epsilon m\right)\mathbb{P}\left(\left|m_y - m/2\right| \geq \epsilon m\right)$$

$$\leq \sum_y 2\exp\left(-\frac{(m_y - 1)m^2\epsilon^2}{32m_y^2} \mid \frac{1}{4}m \leq m_y \leq \frac{3}{4}m\right) + \delta_1$$

$$\leq \sum_y 2 \exp\left(-\frac{(3m/4-1)m^2\epsilon^2}{32(3m/4)^2}\right) + \delta_1 \quad (\frac{x-1}{x^2} \text{ decreases when } x \geq 2)$$

$$\leq 4\exp\left(-\frac{m\epsilon^2}{36}\right) + \delta_1 = \delta_3$$

We can conclude that

$$\mathbb{P}\left(\cup_{i=1}^d \cup_y \frac{|\widehat{\mu_{yi}} - \mu_{yi}|}{\sigma} \geq \epsilon \cup \cup_{i=1}^d \left|\frac{\widehat{\sigma_i^2}}{\sigma^2} - 1\right| \geq \epsilon\right)$$

$$= 2d\delta_2 + d\delta_3$$

$$= 2d\delta_1 + 2d\exp\left(-\frac{m\epsilon^2}{8}\right) + d\delta_1 + 8d\exp\left(-\frac{m\epsilon^2}{36}\right)$$

$$= 6d\exp(-2m\epsilon^2) + 2d\exp\left(-\frac{m\epsilon^2}{8}\right) + 8d\exp\left(-\frac{m\epsilon^2}{36}\right)$$

$$\leq 16d\exp\left(-\frac{m\epsilon^2}{36}\right)$$

Equivalently, when $m = \frac{36}{\epsilon^2}\log\left(\frac{16d}{\delta}\right) = O\left(\frac{1}{\epsilon^2}\log\left(\frac{d}{\delta}\right)\right)$, for any $\delta \in (0,1)$, with probability at least $1 - \delta$, for any $i \in \{1, \dots, d\}$, we have

$$\left|\frac{\widehat{\sigma_i^2}}{\sigma^2} - 1\right| \leq \epsilon, \quad \frac{|\widehat{\mu_{yi}} - \mu_{yi}|}{\sigma} \leq \epsilon,$$

which completes the proof of Lemma B.8. □

Based on the Lemma B.8, we can prove Lemma B.7 as follows.

*Proof.* Without loss of generality, we let $m = O\left(\frac{1}{\epsilon^2}\log\left(\frac{d}{\delta}\right)\right)$ as that in Lemma B.8. We can bound $\mathcal{D}_{\mathrm{KL}}(\mathcal{D}_G(S)\|\mathcal{D})$ as follows.

$$\mathcal{D}_{\mathrm{KL}}(\mathcal{D}_G(S)\|\mathcal{D})$$
$$= \int p_G(\mathbf{x}, y) \log \frac{p_G(\mathbf{x}, y)}{p(\mathbf{x}, y)}$$
$$= \int p_G(\mathbf{x}, y) \log \frac{p_G(\mathbf{x} \mid y)p_G(y)}{p(\mathbf{x} \mid y)p(y)}$$
$$= \int p_G(\mathbf{x}, y) \log \frac{p_G(\mathbf{x} \mid y)}{p(\mathbf{x} \mid y)} \quad (p_G(y) = p(y))$$
$$= \int_y p_G(y) \int_x p_G(\mathbf{x} \mid y) \log \frac{p_G(\mathbf{x} \mid y)}{p(\mathbf{x} \mid y)}$$
$$= \sum_y \frac{1}{2} \int_x p_G(\mathbf{x} \mid y) \log \frac{p_G(\mathbf{x} \mid y)}{p(\mathbf{x} \mid y)}$$

$$= \sum_y \frac{1}{2} \sum_{i=1}^d \frac{1}{2} \left( \frac{\widehat{\sigma_i^2}}{\sigma^2} - 1 - \log \left( \frac{\widehat{\sigma_i^2}}{\sigma^2} \right) + \frac{(\widehat{\mu_{yi}} - \mu_{yi})^2}{\sigma^2} \right)$$

$$\leq \sum_y \frac{1}{2} \sum_{i=1}^d \frac{1}{2} \left( \left( \frac{\widehat{\sigma_i^2}}{\sigma^2} - 1 \right)^2 + \frac{(\widehat{\mu_{yi}} - \mu_{yi})^2}{\sigma^2} \right) \qquad (x - log(x+1) \leq x^2, |x| \leq 1/2)$$

$$\leq \sum_y \frac{1}{2} \sum_{i=1}^d \frac{1}{2} \left( \epsilon^2 + \epsilon^2 \right) = d\epsilon^2 \lesssim \frac{d}{m} \log \left( \frac{d}{\delta} \right). \qquad \text{(Lemma B.8)}$$

Finally, by the Pinsker's inequality (such as, [79]), we have

$$\mathcal{D}_{\text{TV}}(\mathcal{D}, \mathcal{D}_G(S)) \leq \min \left( 1, \sqrt{2 \log 2 \mathcal{D}_{\text{KL}}(\mathcal{D}_G(S), \mathcal{D})} \right) \lesssim \min \left( 1, \sqrt{\frac{d}{m} \log \left( \frac{d}{\delta} \right)} \right),$$

which completes the proof of Lemma B.7. $\qquad \square$

**Lemma B.9** (Concentration bound for $\mathcal{T}(m_S, m_G)$). *Let $\delta$ in Lemma B.7 be $\delta_1$, and $\delta$ in Lemma B.4 be $\delta_2$, then With probability at least $1 - \delta_1 - \delta_2$, it holds that*

$$\mathcal{T}(m_S, m_G) \lesssim \min \left( 1, \frac{\sqrt{m_G d}}{m_S} \log \left( \frac{m_S d}{\delta_2} \right) \right).$$

*Proof.* By the triangle inequality, we have

$$\mathcal{D}_{\text{TV}} \left( \mathcal{D}_G^{m_G}(S), \mathcal{D}_G^{m_G}(S^i) \right) \leq \mathcal{D}_{\text{TV}} \left( \mathcal{D}_G^{m_G}(S), \mathcal{D}_G^{m_G}(S^{\backslash i}) \right) + \mathcal{D}_{\text{TV}} \left( S^{\backslash i}, \mathcal{D}_G^{m_G}(S^i) \right).$$

In order to bound $\mathcal{D}_{\text{TV}} \left( \mathcal{D}_G^{m_G}(S), \mathcal{D}_G^{m_G}(S^i) \right)$, We discuss the concentration property of $\mathcal{D}_{\text{TV}} \left( \mathcal{D}_G^{m_G}(S), \mathcal{D}_G^{m_G}(S^{\backslash i}) \right)$, and the same result will hold for $\mathcal{D}_{\text{TV}} \left( S^{\backslash i}, \mathcal{D}_G^{m_G}(S^i) \right)$. In a similar way as the proof of Lemma B.7, we discuss KL divergence $\mathcal{D}_{\text{KL}} \left( \mathcal{D}_G^{m_G}(S), \mathcal{D}_G^{m_G}(S^{\backslash i}) \right)$ at first.

As stated in Lemma B.8, without loss of generation, we assume that $\epsilon \leq 1/4$, and $m_y$ be the number of samples from category $y$, we have $m_y \geq m/2 - \epsilon m \geq m/4$, $m_y \leq m/2 + \epsilon m \geq 3m/4$, and $\left| \widehat{\sigma_i^2}/\sigma^2 - 1 \right| \leq \epsilon$ with probability at least $1 - \delta_1$.

In addition, by Lemma B.4, given a set $S = \{(\mathbf{x}_1, y_1), \ldots, (\mathbf{x}_m, y_m)\}$ and $\mathbf{z}_i'$ sampled from the binary mixture Gaussian distribution, with probability at least $1 - \delta_2$ we have

$$\max_i \|\mathbf{x}_i\|_2 \lesssim \sigma \sqrt{d + \log(\frac{m+1}{\delta_2})}.$$

Therefore, by the union bound, the above statements hold with high probability at least $1 - \delta_1 - \delta_2$. We use $\widehat{\sigma}_{k,\backslash i}^2$ to denote the $k$th-dimension variance learned on the set $S^{\backslash i}$, and $\widehat{\mu}_{yk,\backslash i}$ to denote the learned $k$th-dimension mean of the class $y$. We can simplify $\mathcal{D}_{\text{KL}} \left( \mathcal{D}_G^{m_G}(S), \mathcal{D}_G^{m_G}(S^{\backslash i}) \right)$ as follows,

$$\mathcal{D}_{\text{KL}} \left( \mathcal{D}_G^{m_G}(S), \mathcal{D}_G^{m_G}(S^{\backslash i}) \right)$$

$$= m_G \mathcal{D}_{\text{KL}} \left( \mathcal{D}_G(S), \mathcal{D}_G(S^{\backslash i}) \right)$$

$$= m_G \int p_G(\mathbf{x}, y) \log \frac{p_G(\mathbf{x}, y)}{p_{G^{\backslash i}}(\mathbf{x}, y)}$$

$$= m_G \int p_G(\mathbf{x}, y) \log \frac{p_G(\mathbf{x} \mid y) p_G(y)}{p_{G^{\backslash i}}(\mathbf{x} \mid y) p_{G^{\backslash i}}(y)}$$

$$
\begin{aligned}
&= m_G \int p_G(\mathbf{x}, y) \log \frac{p_G(\mathbf{x} \mid y)}{p_{G \backslash i}(\mathbf{x} \mid y)} && (p_G(y) = p_{G \backslash i}(y)) \\
&= m_G \int_y p_G(y) \int_x p_G(\mathbf{x} \mid y) \log \frac{p_G(\mathbf{x} \mid y)}{p_{G \backslash i}(\mathbf{x} \mid y)} \\
&= m_G \sum_y \frac{1}{2} \int_x p_G(\mathbf{x} \mid y) \log \frac{p_G(\mathbf{x} \mid y)}{p_{G \backslash i}(\mathbf{x} \mid y)} \\
&= m_G \sum_y \frac{1}{2} \sum_{k=1}^d \frac{1}{2} \left( \frac{\widehat{\sigma}_k^2}{\widehat{\sigma}_{k,\backslash i}^2} - 1 - \log \left( \frac{\widehat{\sigma}_k^2}{\widehat{\sigma}_{k,\backslash i}^2} \right) + \frac{(\widehat{\mu}_{yk} - \widehat{\mu}_{yk,\backslash i})^2}{\widehat{\sigma}_{k,\backslash i}^2} \right) \\
&\leq m_G \sum_y \frac{1}{2} \sum_{k=1}^d \frac{1}{2} \left( \left( \frac{\widehat{\sigma}_k^2}{\widehat{\sigma}_{k,\backslash i}^2} - 1 \right)^2 + \frac{(\widehat{\mu}_{yk} - \widehat{\mu}_{yk,\backslash i})^2}{\widehat{\sigma}_{k,\backslash i}^2} \right) && (x - log(x+1) \leq x^2, |x| \leq 1/2) \\
&= m_G \sum_y \frac{1}{4} \left( \sum_{k=1}^d \left( \frac{\widehat{\sigma}_k^2}{\widehat{\sigma}_{k,\backslash i}^2} - 1 \right)^2 + \sum_{k=1}^d \frac{(\widehat{\mu}_{yk} - \widehat{\mu}_{yk,\backslash i})^2}{\widehat{\sigma}_{k,\backslash i}^2} \right). && (15)
\end{aligned}
$$

What we need to bound is $\left| \widehat{\sigma}_k^2 - \widehat{\sigma}_{k,\backslash i}^2 \right|$ and $\left| \widehat{\mu}_{yk} - \widehat{\mu}_{yk,\backslash i} \right|$. They can be bounded by using the boundedness of the data. Without the loss of generation, we assume that $y_i = 0$, then we have

$$
\begin{aligned}
\left| \widehat{\mu}_{0k} - \widehat{\mu}_{0k,\backslash i} \right| &= \left| \sum_j \frac{x_{jk}}{m_0} - \sum_{j \neq i} \frac{x_{jk}}{m_0 - 1} \right| \\
&\lesssim \left| \sum_j \frac{x_{jk}}{m_0} - \sum_{j \neq i} \frac{x_{jk}}{m_0} \right| \\
&\lesssim \left| \frac{x_{jk}}{m_0} \right| \lesssim \left| \frac{x_{jk}}{m} \right| \\
&\leq \frac{1}{m} \left| \mu_{0k} + \sqrt{2} \sigma \sqrt{\log \left( \frac{(m+1)d}{\delta_2} \right)} \right| \\
&\lesssim \frac{1}{m} \sigma \sqrt{\log \left( \frac{(m+1)d}{\delta_2} \right)},
\end{aligned}
$$
$$
\left| \widehat{\mu}_{1k} - \widehat{\mu}_{1k,\backslash i} \right| = 0.
$$

Therefore, we have

$$
\begin{aligned}
\sum_{i=1}^d \frac{(\widehat{\mu_{0k}} - \widehat{\mu}_{0k,\backslash i})^2}{\widehat{\sigma}_{k,\backslash i}^2} &\lesssim \sum_{k=1}^d \frac{1}{\widehat{\sigma}_{k,\backslash i}^2} \frac{1}{m^2} \sigma^2 \log \left( \frac{(m+1)d}{\delta_2} \right) \\
&\lesssim \sum_{k=1}^d \frac{1}{m^2} \log \left( \frac{(m+1)d}{\delta_2} \right) \lesssim \frac{d}{m^2} \log \left( \frac{(m+1)d}{\delta_2} \right), && (16)
\end{aligned}
$$
$$
\sum_{i=1}^d \frac{(\widehat{\mu_{1k}} - \widehat{\mu}_{1k,\backslash i})^2}{\widehat{\sigma}_{k,\backslash i}^2} = 0. \tag{17}
$$

In terms of $\left| \widehat{\sigma}_k^2 - \widehat{\sigma}_{k,\backslash i}^2 \right|$, we can write

$$
\begin{aligned}
\left| \widehat{\sigma}_k^2 - \widehat{\sigma}_{k,\backslash i}^2 \right| &= \left| \frac{m_0}{m} \frac{\sum_j (x_{jk} - \widehat{\mu}_{0k})^2}{m_0 - 1} - \frac{m_0 - 1}{m} \frac{\sum_{j \neq i} (x_{jk} - \widehat{\mu}_{0k,\backslash i})^2)^2}{m_0 - 2} \right| \\
&\lesssim \left| \frac{m_0}{m} \frac{\sum_j (x_{jk} - \widehat{\mu}_{0k})^2}{m_0 - 1} - \frac{m_0}{m} \frac{\sum_{j \neq i} (x_{jk} - \widehat{\mu}_{0k,\backslash i})^2}{m_0 - 1} \right| \\
&= \frac{m_0}{m(m_0 - 1)} \left| x_{ik}^2 + (m_0 - 1) \widehat{\mu}_{0k,\backslash i}^2 - m_0 \widehat{\mu}_{0k}^2 \right| \\
&\lesssim \frac{m_0}{m(m_0 - 1)} \left| x_{ik}^2 + m_0 \widehat{\mu}_{0k,\backslash i}^2 - m_0 \widehat{\mu}_{0k}^2 \right| \\
&\lesssim \frac{m_0}{m(m_0 - 1)} \left| x_{ik}^2 + m_0 \left( \widehat{\mu}_{0k,\backslash i}^2 - \widehat{\mu}_{0k}^2 \right) \right| \\
&= \frac{m_0}{m(m_0 - 1)} \left| x_{ik}^2 + m_0 \left( \widehat{\mu}_{0k,\backslash i} - \widehat{\mu}_{0k} \right) \left( \widehat{\mu}_{0k,\backslash i} + \widehat{\mu}_{0k} \right) \right| \\
&= \frac{m_0}{m(m_0 - 1)} \left| x_{ik}^2 + m_0 \left( \sum_{j \neq i} \frac{x_{jk}}{m_0 - 1} - \sum_j \frac{x_{jk}}{m_0} \right) \left( \sum_{j \neq i} \frac{x_{jk}}{m_0 - 1} + \sum_j \frac{x_{jk}}{m_0} \right) \right| \\
&\lesssim \frac{m_0}{m(m_0 - 1)} \left| x_{ik}^2 + m_0 \frac{x_{ik}}{m_0} \left( \sum_{j \neq i} \frac{x_{jk}}{m_0 - 1} + \sum_j \frac{x_{jk}}{m_0} \right) \right| \\
&\lesssim \frac{1}{m} \left( \left| x_{ik}^2 \right| + \left| x_{ik} \left( \sum_{j \neq i} \frac{x_{jk}}{m_0 - 1} + \sum_j \frac{x_{jk}}{m_0} \right) \right| \right) \\
&\lesssim \frac{1}{m} \left( \sigma^2 \log \left( \frac{(m+1)d}{\delta_2} \right) + 2\sigma^2 \log \left( \frac{(m+1)d}{\delta_2} \right) \right) \\
&\lesssim \frac{\sigma^2}{m} \log \left( \frac{(m+1)d}{\delta_2} \right).
\end{aligned}
$$

Thus, we can obtain

$$
\begin{aligned}
\sum_{k=1}^d \left( \frac{\widehat{\sigma}_k^2}{\widehat{\sigma}_{k,\backslash i}^2} - 1 \right)^2 &\lesssim \sum_{k=1}^d \frac{\sigma^4}{\widehat{\sigma}_{k,\backslash i}^4 m^2} \log^2 \left( \frac{(m+1)d}{\delta_2} \right) \\
&\lesssim \frac{d}{m^2} \log^2 \left( \frac{(m+1)d}{\delta_2} \right).
\end{aligned}
\tag{18}
$$

By plugin (16), (17) and (18) into (15), we have

$$
\begin{aligned}
\mathcal{D}_{\mathrm{KL}} \left( \mathcal{D}_G^{m_G}(S), \mathcal{D}_G^{m_G}(S^{\backslash i}) \right) &\leq m_G \sum_y \frac{1}{4} \left( \sum_{k=1}^d \left( \frac{\widehat{\sigma}_k^2}{\widehat{\sigma}_{k,\backslash i}^2} - 1 \right)^2 + \sum_{k=1}^d \frac{(\widehat{\mu}_{yk} - \widehat{\mu}_{yk,\backslash i})^2}{\widehat{\sigma}_{k,\backslash i}^2} \right) \\
&\lesssim m_G \frac{d}{m^2} \log^2 \left( \frac{(m+1)d}{\delta_2} \right),
\end{aligned}
$$

which implies

$$
\mathcal{D}_{\mathrm{TV}} \left( \mathcal{D}_G^{m_G}(S), \mathcal{D}_G^{m_G}(S^{\backslash i}) \right) \lesssim \min \left( 2, \sqrt{\mathcal{D}_{\mathrm{KL}} \left( \mathcal{D}_G^{m_G}(S), \mathcal{D}_G^{m_G}(S^{\backslash i}) \right)} \right)
$$

$$\lesssim \min\left(2, \sqrt{\mathcal{D}_{\mathrm{KL}}\left(\mathcal{D}_G^{m_G}(S), \mathcal{D}_G^{m_G}(S^{\backslash i})\right)}\right)$$

$$\lesssim \min\left(2, \frac{\sqrt{m_G d}}{m}\log\left(\frac{(m+1)d}{\delta_2}\right)\right),$$

and

$$\mathcal{D}_{\mathrm{TV}}\left(\mathcal{D}_G^{m_G}(S), \mathcal{D}_G^{m_G}(S^i)\right) \leq \mathcal{D}_{\mathrm{TV}}\left(\mathcal{D}_G^{m_G}(S), \mathcal{D}_G^{m_G}(S^{\backslash i})\right) + \mathcal{D}_{\mathrm{TV}}\left(S^{\backslash i}, \mathcal{D}_G^{m_G}(S^i)\right)$$

$$\lesssim \max\left(1, \frac{\sqrt{m_G d}}{m}\log\left(\frac{(m+1)d}{\delta_2}\right)\right)$$

$$\lesssim \max\left(1, \frac{\sqrt{m_G d}}{m}\log\left(\frac{md}{\delta_2}\right)\right).$$

Because it holds for all $i$, the proof of Lemma B.9 is completed. $\square$

Now we are ready to prove Theorem 3.2.

*Proof.* Let $\delta$ in Lemma B.7 be $\delta_1$ and that in Lemma B.4 be $\delta_2$. With probability at least $1 - \delta/2$, the bounds in Lemma B.7 hold with $\delta_1 = \delta/2$. Then with probability at least $1 - \delta/2$. Besides, the bounds in Lemma B.5 and Lemma B.6 hold with $\delta_2 = \delta/2$. Thus, by the union bound, we know that with high probability $1 - \delta$, the above bounds hold. Furthermore, from the proof of Lemma B.9, we know that it holds naturally in this case, where the boundedness of data points and the accurate estimation of the true distribution hold.

Finally, we plugin Lemma B.5, B.7, B.6, B.9 into Theorem 3.1, and can conclude the statement of Theorem 3.2 with high probability at least $1 - \delta$,

$$|\textit{Gen-error}|$$

$$\lesssim \frac{m_G}{m_T}\left(d + \log\left(\frac{m_S}{\delta}\right)\right)\min\left(1, \sqrt{\frac{d}{m_S}\log\left(\frac{d}{\delta}\right)}\right)$$

$$+ \frac{\sqrt{m_S} + \sqrt{m_G}}{m_T}\left(d + \log\left(\frac{m_S}{\delta}\right)\right)\sqrt{\log\left(\frac{1}{\delta}\right)} + \frac{m_S\sqrt{m_G}}{m_T^2}\left(d + \log\left(\frac{m_T}{\delta}\right)\right)\sqrt{\log\left(\frac{1}{\delta}\right)}$$

$$+ \frac{m_S\log m_S + m_G\log m_G}{m_T^2}\left(d + \log\left(\frac{m_T}{\delta}\right)\right)\log\left(\frac{1}{\delta}\right)$$

$$+ \frac{m_S\log m_S}{m_T}\left(d + \log\left(\frac{m_S}{\delta}\right)\right)\min\left(1, \frac{\sqrt{m_G d}}{m_S}\log\left(\frac{m_S d}{\delta}\right)\right)\log\left(\frac{1}{\delta}\right) \qquad (19)$$

$$\sim \begin{cases} \frac{\log(m_S)}{\sqrt{m_S}} & \text{if fix } d \text{ and } m_G = 0, \\ \frac{\log^2(m_S)}{\sqrt{m_S}} & \text{if fix } d \text{ and } m_G = \Theta(m_S), \\ \frac{\log(m_S)}{\sqrt{m_S}} & \text{if fix } d \text{ and } m_G = m_{G,\text{order}}^*, \\ d & \text{if fix } m_S. \end{cases}$$

$\square$

**Corollary B.1.** *We denote the generalization error upper bound (Equation (19)) by $Error(m_G)$, where $m_G$ is the augmentation size. We compare the cases where $m_G = 0$ (without GDA) and $m_G \to +\infty$. If $d > m_S$, then the following holds:*

$$Error(+\infty) \leq \frac{1}{\log(m_S)}Error(0).$$

*Proof.* By Equation (19), when $d > m_S$, we have

$$Error(0) = \left( \frac{1}{\sqrt{m_S}} \sqrt{\log\left(\frac{1}{\delta}\right)} + \frac{\log(m_S)}{m_S} \log\left(\frac{1}{\delta}\right) + \log m_S \log\left(\frac{1}{\delta}\right) \right) \left( d + \log\left(\frac{m_S}{\delta}\right) \right),$$

and

$$Error(+\infty) = d + \log\left(\frac{m_S}{\delta}\right).$$

When $\delta$ is sufficiently small, we have

$$Error(0) = \left( \frac{1}{\sqrt{m_S}} \sqrt{\log\left(\frac{1}{\delta}\right)} + \frac{\log(m_S)}{m_S} \log\left(\frac{1}{\delta}\right) + \log m_S \log\left(\frac{1}{\delta}\right) \right) Error(+\infty)$$

$$\geq \log m_S Error(+\infty),$$

which completes the proof. $\qquad\square$

### B.3 Proof of Theorem 3.3

The theorem is built upon the recent theoretical works on GAN [49] and SGD [35; 50]. We first list some lemmas from these works.

**Lemma B.10** (Upper bounds for output and gradient, Proposition 5.2, [50]). *For deep CNNs or MLPs in Appendix A.1, we have*

$$|f(\mathbf{w}, \mathbf{x})| \leq \left( \prod_{l=1}^{L} \|\mathbf{w}_l\|_2 \right) \|\mathbf{x}\|_2,$$

$$\left\| \frac{\partial f(\mathbf{w}, \mathbf{x})}{\partial \mathbf{w}_l} \right\|_2 \leq \left( \prod_{i \neq l} \|\mathbf{w}_l\|_2 \right) \|\mathbf{x}\|_2.$$

**Lemma B.11** (Uniform stability of SGD in the non-convex case, Theorem 5, [35]). *Assume $f$ is $\beta$-smooth and $\rho$-Lipschitz. Running $T > m$ iterations of SGD with step size $\alpha_t = \frac{c}{\beta t}$, the stability of SGD satisfies*

$$\beta_m \leq \frac{16\rho^2 T^c}{m^{1+c}}.$$

**Lemma B.12** (Learnability of GAN, Theorem 19, [49]). *We suppose that the architecture of GAN is the same as that in Appendix A.2. Besides, we consider the realizable setting, that is, $\mathcal{D}$ enjoys the same distribution as $g_{\theta_*}(Z)$ with some $\theta_* \in \Theta(d, L)$ and $Z \sim \text{unif}[0, 1]^d$. Then, given training set $S$ with $m$ i.i.d. samples, it holds that*

$$\mathbb{E}\mathcal{D}_{\text{TV}}^2\left( \mathcal{D}, \mathcal{D}_G(S) \right) \lesssim \sqrt{d^2 L^2 \log(dL) \frac{\log m}{m}}.$$

*Proof.* Now we are ready to prove Theorem 3.3, the main idea is to bound $M$, $\beta_{m_T}$, $\mathcal{D}_{\text{TV}}\left( \mathcal{D}, \mathcal{D}_G(S) \right)$ in Theorem 3.1. $M$ and Lipschitz property can be bounded by using Lemma B.10. $\beta_{m_T}$ can be induced by Lemma B.11 with Lipschitz constant. In terms of $\mathcal{D}_{\text{TV}}\left( \mathcal{D}, \mathcal{D}_G(S) \right)$, Lemma B.12 can be used to derive an upper bound.

First, we bound the loss function as follows.

$$\begin{aligned}
\ell(f, \mathbf{z}) &= \ell(f, (\mathbf{x}, y)) \\
&= \log(1 + \exp(-yf(\mathbf{w}, \mathbf{x}))) \\
&\leq \log(2) + |yf(\mathbf{w}, \mathbf{x})| \qquad\qquad (\log(1 + \exp(-t)) \text{ is 1-Lipschitz}) \\
&= \log(2) + |f(\mathbf{w}, \mathbf{x})|
\end{aligned}$$

$$\leq \log(2) + \left(\prod_{l=1}^{L} \|\mathbf{w}_l\|_2\right) \|\mathbf{x}\|_2 \qquad \text{(by Lemma B.10)}$$

$$\leq \log(2) + \left(\prod_{l=1}^{L} \|\mathbf{w}_l\|_2\right) \sqrt{d}$$

$$\leq \log(2) + \left(\prod_{l=1}^{L} \|W_l\|_2\right) \sqrt{d}$$

$$\lesssim \left(\prod_{l=1}^{L} \|W_l\|_2\right) \sqrt{d}.$$

Thus, we have $M \lesssim \left(\prod_{l=1}^{L} \|W_l\|_2\right) \sqrt{d}$.

Second, we prove that $f$ is Lipschitz given the bounded parameter space.

$$\left\|\frac{\partial f(\mathbf{w}, \mathbf{x})}{\partial \mathbf{w}}\right\|_2 \leq \sum_{l=1}^{L} \left\|\frac{\partial f(\mathbf{w}, \mathbf{x})}{\partial \mathbf{w}_l}\right\|_2$$

$$\leq \|\mathbf{x}\|_2 \sum_{l=1}^{L} \left(\prod_{i \neq l} \|\mathbf{w}_l\|_2\right) \qquad \text{(by Lemma B.10)}$$

$$\leq \|\mathbf{x}\|_2 \sum_{l=1}^{L} \left(\prod_{i \neq l} \|W_l\|_2\right)$$

$$\leq \sqrt{d} \sum_{l=1}^{L} \left(\prod_{i \neq l} \|\mathbf{w}_l\|_2\right)$$

$$\lesssim \sqrt{d} \sum_{l=1}^{L} \left(\prod_{i} \|W_l\|_2\right)$$

$$= \sqrt{d}L \left(\prod_{i} \|W_l\|_2\right).$$

Therefore, $f$ is $\rho$-Lipschitz with $\sqrt{d}L\left(\prod_i \|W_l\|_2\right)$. Then, $\beta_m$ can be bounded by Lemma B.11.

$$\beta_m \leq \frac{16\rho^2 T^c}{m^{1+c}} \leq 16dL^2 \left(\prod_i \|W_l\|_2\right)^2 \frac{T^c}{m^{1+c}} \lesssim \left(\prod_i \|W_l\|_2\right)^2 \frac{dL^2}{m}.$$

Third, we bound the expectation of divergence between model distribution and target distribution as follows.

$$\mathbb{E}\mathcal{D}_{\mathrm{TV}}\left(\mathcal{D}, \mathcal{D}_G(S)\right) \lesssim \mathbb{E} \int_{(\mathbf{x}, y)} \left|\mathbb{P}_{\mathcal{D}}(\mathbf{x}, y) - \mathbb{P}_{\mathcal{D}_G(S)}(\mathbf{x}, y)\right| d\mathbf{z}$$

$$= \mathbb{E} \sum_{y} \int_{\mathbf{x}} \left|\mathbb{P}_{\mathcal{D}}(\mathbf{x}, y) - \mathbb{P}_{\mathcal{D}_G(S)}(\mathbf{x}, y)\right| d\mathbf{x}$$

$$= \mathbb{E} \sum_{y} \int_{\mathbf{x}} \frac{1}{2} \left|\mathbb{P}_{\mathcal{D}}(\mathbf{x} \mid y) - \mathbb{P}_{\mathcal{D}_G(S)}(\mathbf{x} \mid y)\right| d\mathbf{x}$$

$$= \mathbb{E} \sum_y \mathcal{D}_{\mathrm{TV}} \left( \mathbb{P}_{\mathcal{D}}(\mathbf{x} \mid y), \mathbb{P}_{\mathcal{D}_G(S)}(\mathbf{x} \mid y) \right)$$

$$= \sum_y \mathbb{E} \mathcal{D}_{\mathrm{TV}} \left( \mathbb{P}_{\mathcal{D}}(\mathbf{x} \mid y), \mathbb{P}_{\mathcal{D}_G(S)}(\mathbf{x} \mid y) \right)$$

$$= \sum_y \sqrt{\left( \mathbb{E} \mathcal{D}_{\mathrm{TV}} \left( \mathbb{P}_{\mathcal{D}}(\mathbf{x} \mid y), \mathbb{P}_{\mathcal{D}_G(S)}(\mathbf{x} \mid y) \right) \right)^2}$$

$$\leq \sum_y \sqrt{\mathbb{E} \mathcal{D}_{\mathrm{TV}}^2 \left( \mathbb{P}_{\mathcal{D}}(\mathbf{x} \mid y), \mathbb{P}_{\mathcal{D}_G(S)}(\mathbf{x} \mid y) \right)}$$

$$\lesssim \sum_y \sqrt{\sqrt{d^2 L^2 \log(dL) \frac{\log m}{m}}}$$

$$\lesssim \left( d^2 L^2 \log(dL) \frac{\log m}{m} \right)^{\frac{1}{4}} = \sqrt{dL} \left( \log(dL) \frac{\log m}{m} \right)^{\frac{1}{4}}.$$

Furthermore, because $\mathcal{D}_{\mathrm{TV}} \left( \mathcal{D}, \mathcal{D}_G(S) \right) \leq 1$, we have

$$\mathbb{E} \mathcal{D}_{\mathrm{TV}} \left( \mathcal{D}, \mathcal{D}_G(S) \right) \lesssim \min \left( 1, \sqrt{dL} \left( \log(dL) \frac{\log m}{m} \right)^{\frac{1}{4}} \right).$$

Finally, by taking the expectation for the bound in Theorem 3.1, and plugging $M$, $\beta_{m_T}$ and $\mathcal{D}_{\mathrm{TV}} \left( \mathcal{D}, \mathcal{D}_G(S) \right)$ into it, we can conclude the result of Theorem 3.3 with high probability at least $1 - \delta$,

$M \lesssim \left( \prod_{l=1}^L \|W_l\|_2 \right) \sqrt{d}$

$$\beta_m \leq \frac{16\rho^2 T^c}{m^{1+c}} \leq 16dL^2 \left( \prod_i \|W_l\|_2 \right)^2 \frac{T^c}{m^{1+c}} \lesssim \left( \prod_i \|W_l\|_2 \right)^2 \frac{dL^2}{m}.$$

$\mathbb{E}|Gen\text{-}error|$

$$\lesssim \frac{m_G}{m_T} \left( \prod_{l=1}^L \|W_l\|_2 \right) \sqrt{d} \min \left( 1, \sqrt{dL} \left( \log(dL) \frac{\log m_S}{m_S} \right)^{\frac{1}{4}} \right)$$

$$+ \frac{\sqrt{m_S} + \sqrt{m_G}}{m_T} \left( \prod_{l=1}^L \|W_l\|_2 \right) \sqrt{d} \sqrt{\log \left( \frac{1}{\delta} \right)} + \frac{m_S \sqrt{m_G}}{m_T^2} \left( \prod_i \|W_l\|_2 \right)^2 dL^2 \sqrt{\log \left( \frac{1}{\delta} \right)}$$

$$+ \frac{m_S \log m_S + m_G \log m_G}{m_T^2} \left( \prod_i \|W_l\|_2 \right)^2 dL^2 \log \left( \frac{1}{\delta} \right)$$

$$+ \frac{m_S \log m_S}{m_T} \left( \prod_{l=1}^L \|W_l\|_2 \right) \sqrt{d} \mathcal{T}(m_S, m_G) \log \left( \frac{1}{\delta} \right) \tag{20}$$

$$
\lesssim
\begin{cases}
\frac{1}{\sqrt{m_S}} & \text{if fix } W, L, d, \text{ let } m_G = 0, \\[2mm]
\max\left( \left(\frac{\log(m_S)}{m_S}\right)^{\frac{1}{4}}, \log m_S \mathcal{T}(m_S, m_G) \right) & \text{if fix } W, L, d, \text{ let } m_G = \Theta(m_S), \\[2mm]
\left(\frac{\log(m_S)}{m_S}\right)^{\frac{1}{4}} & \text{if fix } W, L, d, \text{ let } m_G = m^*_{G,\text{order}}, \\[2mm]
dL^2 \left(\prod_{l=1}^{L} \|W_l\|_2\right)^2 & \text{if fix } m_S.
\end{cases}
$$

$\square$

**Corollary B.2.** *We denote the generalization error upper bound (Equation (20)) by $Error(m_G)$, where $m_G$ is the augmentation size. We compare the cases where $m_G = 0$ (without GDA) and $m_G \to +\infty$. If $d > m_S^2$, then the following holds:*

$$
Error(+\infty) \leq \frac{1}{\prod_{l=1}^{L} \|W_l\|_2 \, L^2} Error(0).
$$

*Proof.* By Equation (20), when $d > m_S^2$, we have

$$
Error(0) = \frac{1}{\sqrt{m_S}} \left(\prod_{l=1}^{L} \|W_l\|_2\right) \sqrt{d}\sqrt{\log\left(\frac{1}{\delta}\right)} + \frac{\log m_S}{m_S} \left(\prod_i \|W_l\|_2\right)^2 dL^2 \log\left(\frac{1}{\delta}\right),
$$

and

$$
Error(+\infty) = \left(\prod_{l=1}^{L} \|W_l\|_2\right) \sqrt{d}.
$$

When $\delta$ is sufficiently small, we have

$$
Error(0)
$$
$$
= \frac{1}{\sqrt{m_S}} Error(+\infty)\sqrt{\log\left(\frac{1}{\delta}\right)} + \frac{\log m_S}{m_S} \left(\prod_i \|W_l\|_2\right) \sqrt{d}L^2 Error(+\infty) \log\left(\frac{1}{\delta}\right)
$$
$$
\geq \frac{\log m_S}{m_S} \left(\prod_i \|W_l\|_2\right) \sqrt{d}L^2 Error(+\infty)
$$
$$
\geq \frac{1}{m_S} \left(\prod_i \|W_l\|_2\right) \sqrt{d}L^2 Error(+\infty)
$$
$$
\geq \left(\prod_i \|W_l\|_2\right) L^2 Error(+\infty),
$$

which completes the proof. $\square$

## Appendix C   Discussion on existing non-i.i.d. stability bounds

In this section, we show that it is unclear how to use existing non-i.i.d. stability bounds to derive a better guarantee than Theorem 3.1 for GDA.

### C.1   Stability bounds for mixing processes

To the best of our knowledge, existing stability bounds for mixing processes only focus on the stationary sequence [27; 28; 72], which is defined as follows.

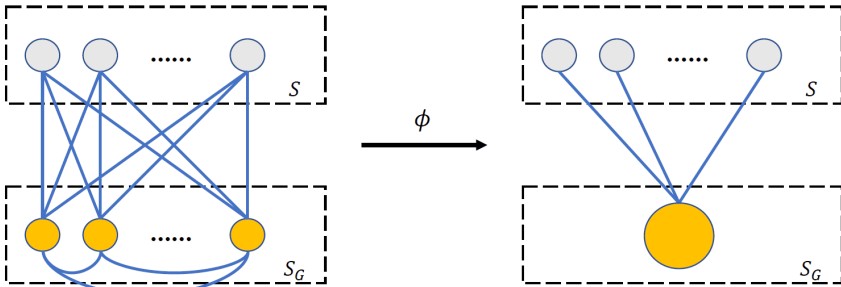

Figure 2: Dependence graph (left) and a forest approximation (right) of the GDA setting.

**Definition C.1** (Stationary sequence). *A sequence of random variables $\mathbf{Z} = \{Z_t\}_{t=-\infty}^{\infty}$ is said to be stationary if for any $t$ and non-negative integers $m$ and $k$, the random vectors $(Z_t, \ldots, Z_{t+m})$ and $(Z_{t+k}, \ldots, Z_{t+m+k})$ have the same distribution.*

Unfortunately, the GDA setting in this paper does not satisfy the stationary condition, because $(\mathbf{z}_1, \ldots, \mathbf{z}_{m_S}) = S$ and $(\mathbf{z}_{m_S+1}, \ldots, \mathbf{z}_{2m_S}) \subseteq S_G$ do not have the same distribution. Furthermore, it is usually difficult to estimate the mixing coefficients which reflect quantitative dependencies among data points.

### C.2 Stability bounds for dependence graph

Recently, [29] provide a framework for the generalization theory of graph-dependent data, which includes the classical stability result in [25] as a special case. We now introduce some elements of graph-dependent random variables and the non-i.i.d. stability bound in [29]. For a graph $G$, we use $V(G)$ to denote its vertex set and $E(G)$ to denote its edge set.

**Definition C.2** (Dependency Graph, Definition 3.1 in [29]). *An undirected graph $G$ is called a dependency graph of a random vector $\mathbf{X} = (X_1, \ldots, X_n)$ if (1) $V(G) = [n]$, (2) if $I, J \subseteq [n]$ are non-adjacent in $G$, then $\{X_i\}_{i \in I}$ and $\{X_j\}_{j \in J}$ are independent.*

**Definition C.3** (Forest Approximation, Definition 3.4 in [29]). *Given a graph $G$, a forest $F$, and a mapping $\phi : V(G) \rightarrow V(F)$, if $\phi(u) = \phi(v)$ or edge $\langle \phi(u), \phi(v) \rangle \in E(F)$ for any edge $\langle \phi(u), \phi(v) \rangle \in E(G)$, then $(\phi, F)$ is called a forest approximation of $G$. Let $\Phi(G)$ be the set of forest approximations of $G$.*

**Definition C.4** (Forest Complexity, Definition 3.5 in [29]). *Given a graph $G$ and any forest approximation $(\phi, F) \in \Phi(G)$ with $F$ consisting of trees $\{T_i\}_{i \in [k]}$, let*

$$\lambda_{(\phi,F)} = \sum_{\langle u,v \rangle \in E(F)} \left( \left| \phi^{-1}(u) \right| + \left| \phi^{-1}(v) \right| \right)^2 + \sum_{i=1}^{k} \min_{u \in V(T_i)} \left| \phi^{-1}(u) \right|^2 .$$

*We call $\Lambda(G) = \min_{(\phi,F) \in \Phi(G)} \lambda_{(\phi,F)}$ the forest complexity of the graph $G$.*

**Theorem C.1.** *Assume that $\mathcal{A}$ is a $\beta_m$-stable. Given a set $\widetilde{S}$ of size $m$ sampled from the same marginal distribution $\mathcal{D}$ with dependency graph $G$. Suppose the maximum degree of $G$ is $\Delta$, and the loss function $\ell$ is bounded by $M$. For any $\delta \in (0, 1)$, with probability at least $1 - \delta$, it holds that*

$$\mathcal{R}_{\mathcal{D}}(\mathcal{A}(\widetilde{S})) \leq \widehat{\mathcal{R}}_{\widetilde{S}}(\mathcal{A}(\widetilde{S})) + 2\beta_{m,\Delta}(\Delta + 1) + (4\beta_m + \frac{M}{m})\sqrt{\frac{\Lambda(G)}{2}\log(\frac{1}{\delta})},$$

*where $\beta_{m,\Delta} = \max_{i \leq \Delta} \beta_{m-i}$ and $\Lambda(G)$ is the forest complexity of the dependence graph $G$.*

*Remark.* Theorem C.1 requires $\widetilde{S}$ sampled from the same marginal distribution $\mathcal{D}$, which fails to hold in the context of GDA because the learned distribution $\mathcal{D}_G(S)$ is generally not the same as the true distribution $\mathcal{D}$. It is still unclear to overcome this problem.

*Remark.* When $m_G = 0$ and $\widetilde{S} = S$, Theorem C.1 degenerates to the classical result in [25], which requires $\beta_m = o(1/\sqrt{m})$ to converge. In contrast, Theorem 3.1 only requires $\beta_m = o(1/\log(m))$ to converge, which is better than that of Theorem C.1.

*Remark.* We note that Theorem C.1 is proposed for the general case with data dependence. Therefore, it does not consider the property of special cases and may fail to give good guarantees. On the one hand, the independence of $S$ and the conditional independence of $S_G$ used in the proof of Theorem 3.1 are significant, which is ignored by Theorem C.1. On the other hand, in the case of strong dependence like GDA, the forest complexity may be too large to give a meaningful bound. The dependence graph and a forest approximation of the GDA setting are presented in Figure 2. Therefore, the forest complexity of the GDA setting can be bounded as follows.

$$\Lambda(G) \le m_S(1 + m_G)^2 + 1^2 \lesssim m_S m_G^2. \tag{21}$$

Plugging (21) into Theorem C.1, and assume $m_G = \Theta(m_S)$, we observe that

$$\frac{M}{m_T} \sqrt{\frac{\Lambda(G)}{2} \log(\frac{1}{\delta})} \lesssim \frac{M}{m_T} \sqrt{\frac{m_S m_G^2}{2} \log(\frac{1}{\delta})} \lesssim M \sqrt{\frac{m_S}{2} \log(\frac{1}{\delta})},$$

which fails to converge. However, Theorem 3.1 overcomes this problem.

Finally, we conclude that it is hard to directly use existing non-i.i.d. stability results to obtain a better guarantee than Theorem 3.1.

## Appendix D   Experimental details and additional results

### D.1   CIFAR-10 dataset

CIFAR-10 is a widely used image dataset and we adopt it to empirically validate Theorem 3.3. Combining the simulations in the bGMM setting, our theory is verified sufficiently.

### D.2   Models

**bGMM**. We adopt the implementation of naïve Bayes in [80] to estimate the parameters of bGMM.

**ResNet.** We add the ResNet50 checkpoint released by Pytorch [81], which is also used in [24].

**cDCGAN.** We use the cDCGAN in this repository, and modify its input channel and label dimension to 3 and 10 respectively to keep consistent with the format of images in CIFAR-10 dataset. This repository gains the most stars among repositories that implement cDCGAN. Furthermore, we follow its hyperparameter setting and train 200 epochs to obtain a cDCGAN for the CIFAR-10 dataset.

**StyleGAN2-ADA.** We use the class-conditional model pre-trained on CIFAR-10 dataset, which is released by NVIDIA Research [56].

**EDM.** We use the 5M synthetic CIFAR-10 dataset released in [24], which is generated by the pre-trained conditional EDM. Given an augmentation size $m_G$, we randomly sample $m_G$ from the 5M synthetic data points.

### D.3   Model selection

GANs are chosen to empirically validate Theorem 3.3 and the EDM is chosen to explore the ability of the diffusion model. First, we choose a "bad" GAN (DCGAN) to empirically verify that GANs can improve the test performance when $m_S$ is small and awful overfitting happens (without standard augmentation). Second, we choose a "good" GAN (StyleGAN2-ADA) to verify that GANs can not improve the test performance obviously when the $m_S$ is approximately large (with standard augmentation). Third, because diffusion models have achieved good success in recent years, we conduct experiments on the EDM and suggest that diffusion models have a better $\mathcal{D}_{\mathrm{TV}}(\mathcal{D}, \mathcal{D}_G(S))$ than GANs.

### D.4   Training details

**Standard data augmentation.** 4 pixels are padded on each side, and a $32 \times 32$ crop is randomly sampled from the padded image or its horizontal flip. This augmentation pipeline is widely used [54].

**Optimization.** We follow the setting in [24]. We use the SGD optimizer, where the momentum and weight decay are set to 0.9 and $5 \times 10^{-4}$, respectively. We use the cyclic learning rate schedule with cosine annealing, where the initial learning rate is set to 0.2. We train the deep neural classifier with 100 epochs. The batch size is 512.

## D.5 Computation consumption.

All experiments are run on one RTX 3090 GPU. The most consuming case (ResNet50, $m_G = 1M$) takes 17 GB cuda memory and 20 hours.

## D.6 License

The used codes and their licenses are listed in Table 2.

Table 2: The used codes and licenses.

| URL | Citation | License |
|---|---|---|
| https://github.com/NVlabs/stylegan2-ada-pytorch | [56] | License |
| https://github.com/pytorch/pytorch | [81] | License |
| https://github.com/wzekai99/DM-Improves-AT | [24] | MIT License |
| https://github.com/ML-GSAI/Revisiting-Dis-vs-Gen-Classifiers | [80] | MIT License |
| https://github.com/znxlwm/pytorch-MNIST-CelebA-cGAN-cDCGAN | - | - |

## D.7 Additional results

We further adopt the CIFAR-10 dataset to empirically verify our theory by estimating the generalization error directly. By definition, given a trained neural classifier, the generalization error of Theorem 3.3 can be estimated by the absolute gap between the mean cross-entropy loss on the training set (with generated data) and the mean cross-entropy loss on the test set. We add the results of cDCGAN, StyleGAN2-ADA, and EDM with this estimator in Table 3.

On the one hand, GANs decrease the generalization error when $m_S$ is small (without standard augmentation). On the other hand, GANs fail to boost the performance obviously and even hurt the error when $m_S$ is approximately large (with standard augmentation). Our experimental results support the theoretical results (Theorem 3.3) again.

Table 3: Estimated generalization error on the CIFAR-10 dataset, where S.A. denotes standard augmentation.

| Generator | Classifier | S.A. | GDA ($m_G$) | | | | | |
|-----------|-----------|------|------|------|------|------|------|------|
| | | | w/o | 100k | 300k | 500k | 700k | 1M |
| cDCGAN | ResNet18 | × | 0.476 | 0.456 | 0.413 | 0.424 | 0.428 | 0.455 |
| | | √ | 0.227 | 0.227 | 0.227 | 0.221 | 0.238 | 0.240 |
| | ResNet34 | × | 0.538 | 0.564 | 0.476 | 0.474 | 0.514 | 0.514 |
| | | √ | 0.219 | 0.234 | 0.223 | 0.231 | 0.239 | 0.247 |
| | ResNet50 | × | 0.634 | 0.496 | 0.471 | 0.531 | 0.533 | 0.566 |
| | | √ | 0.235 | 0.231 | 0.244 | 0.234 | 0.254 | 0.266 |
| StyleGAN2-ADA | ResNet18 | × | 0.476 | 0.336 | 0.296 | 0.298 | 0.292 | 0.303 |
| | | √ | 0.227 | 0.205 | 0.215 | 0.205 | 0.210 | 0.210 |
| | ResNet34 | × | 0.538 | 0.381 | 0.340 | 0.335 | 0.339 | 0.346 |
| | | √ | 0.219 | 0.222 | 0.219 | 0.236 | 0.229 | 0.223 |
| | ResNet50 | × | 0.634 | 0.357 | 0.313 | 0.330 | 0.331 | 0.322 |
| | | √ | 0.235 | 0.236 | 0.211 | 0.223 | 0.198 | 0.223 |
| EDM | ResNet18 | × | 0.476 | 0.249 | 0.185 | 0.172 | 0.154 | 0.142 |
| | | √ | 0.227 | 0.159 | 0.121 | 0.100 | 0.090 | 0.070 |
| | ResNet34 | × | 0.538 | 0.281 | 0.215 | 0.183 | 0.163 | 0.150 |
| | | √ | 0.219 | 0.164 | 0.120 | 0.100 | 0.096 | 0.084 |
| | ResNet50 | × | 0.634 | 0.265 | 0.194 | 0.186 | 0.172 | 0.149 |
| | | √ | 0.235 | 0.160 | 0.121 | 0.101 | 0.089 | 0.078 |