# OpenReview forum: "Toward Understanding Generative Data Augmentation"
_NeurIPS.cc/2023/Conference — NeurIPS 2023 poster_

### Official Review · Reviewer_sAdG · 2023-06-26

**Soundness:** 3 good
**Presentation:** 4 excellent
**Contribution:** 3 good
**Rating:** 6
**Confidence:** 3

**Summary:**

The paper presents a theoretical study of the generalization properties of training when the training set is augmented with artificially generated data. The main result of the paper is a theorem which bounds the generalization error by two terms – one representing the divergence between the original training distribution and the augmented distribution, and one representing the generalization error of the mixed distribution. The authors presents two empirical contributions. Firstly they study a gaussian mixture model with synthetic data and find that their theoretical predictions match the measurements. Secondly they consider Resnets trained on cifar and find that generative models are useful when augmentations are not used, and that diffusion models but not GANs are useful with augmentations.


EDIT: increased confidence from 2 to 3 after clarifications from the authors.

**Strengths:**


* The topic is topical and potentially impactful.
* The writing is very good.
* The results of the theorem are natural and intuitive.


**Weaknesses:**

* It seems like a major assumption is that “because the distribution learned by the generative model is dependent on the sampled train set”. In practice, I don’t think this is necessarily true. If this assumption is removed, the theoretical results might be much easier to derive.
* Gaussian mixture models are not really used in practice, so it’s not a very interesting experiment.
* It doesn’t seem like the theoretical results will apply to deep neural networks.


**Questions:**

1/ Can you comment on the assumption that  “ the distribution learned by the generative model is dependent on the sampled train set”. Is it realistic? What happens if it is removed?
2. How does the experiments on cifar10 related to your theoretical results?
3. How well does your theoretical results apply to deep neural networks?

---

> ### Author Rebuttal · Authors · 2023-08-09
>
> # Response to Reviewer sAdG
>
> We thank Reviewer sAdG for the positive score and valuable comments.
>
> ## Weakness 1 & Q1:  Assumption "distribution learned by the generative model is dependent on the sampled train set”
>
> Thanks for the suggestion. **This is not an assumption but a main challenge we have to face to establish a theory that is conformed to reality**. This statement holds naturally because even when we change one data point of the input, a learning algorithm may output a very different result. The reviewer can refer to the algorithmic stability theory we summarized in the related work.
>
> ## Weakness 2: bGMM experiment is not interesting
>
> Thanks for the advice. **The bGMM experiments are conducted to clarify our theoretical results (Theorem 3.2) clearly and verify our framework (Theorem 3.1)**, rather than approximating the real-world problem. As a supplement, we design experiments on the CIFAR-10 dataset to show our implications on the practice.
>
> ## Q2: How do the experiments on cifar10 related to the theoretical results
>
> Thanks for the suggestion. The experiments conducted on the CIFAR-10 are used to verify Theorem 3.3.
>
> **The theoretical implications on the deep neural networks (Theorem 3.3) can be summarized as follows.**
>
> * When $m_S$ is large enough, it is hard to boost the performance by augmenting the train set based on GANs. GDA may even damage the generalization. (Remark on line 276)
> * When $m_S$ is small and awful overfitting happens, GANs can improve the test performance. (Remark on line 280)
>
> **To verify our results (Theorem 3.3)**, we designed the experiments on the CIFAR-10 dataset as follows.
>
> * When the $m_S$ is approximately large (with standard augmentation), we choose a "good" GAN (StyleGANv2) to verify that it is hard to use GANs to improve the test performance.
> * When $m_S$ is small and awful overfitting happens (without standard augmentation), we choose a "bad" GAN (DCGAN) to verify that GANs can improve the test performance.
>
> The analysis of the experimental results can be found in the next section. We will clarify these more clearly in the final version.
>
> ## Weakness 3 & Q3: How well do the theoretical results apply to deep neural networks
>
> Thanks for the suggestion. Theorem 3.1 establishes a general framework to understand the GDA, and Theorem 3.3 particularizes it to deep neural networks (GANs). **The experimental results (please see Table 2) support our theory on the deep neural networks (Theorem 3.3)**, which is agreed by Reviewer SG4V and jB6r. The experimental results can be divided into two parts as follows.
>
> * GANs improve the test performance of classifiers when $m_S$ is small  (without standard augmentation) and awful overfitting occurs, though DCGAN can not generate high-quality images. This supports the Remark on line 280.
> * GANs can not boost the performance obviously and even damage it when $m_S$ is approximately large (with standard augmentation). GDA with DCGAN always hurts the generalization ability. Though we use StyleGAN2-ADA (a state-of-the-art GAN), we can not boost the performance of classifiers obviously, and even consistently obtain worse test accuracy when $m_G$ is 500k or 1M.  This supports the Remark on line 276.
>
> Therefore, our experiments support that the theoretical results apply well to deep neural networks. We will discuss these in detail in the final version.

---

> > ### Comment · Reviewer_sAdG · 2023-08-12
> > **Thanks for your reply**
> >
> > Thanks for your reply. I'm happy for the clarifications regarding how the theory applies to ANNs. I will keep my score but will increase my confidence.

---

> > > ### Author Response · Authors · 2023-08-13
> > >
> > > Thanks again for your valuable comments and acknowledgment of our work.

---

### Official Review · Reviewer_hFUC · 2023-06-27

**Soundness:** 3 good
**Presentation:** 2 fair
**Contribution:** 2 fair
**Rating:** 5
**Confidence:** 2

**Summary:**

The paper provides a theoretical analysis of the stability bound for generative data augmentation. The authors provide empirical evidence to validate the proposed theory on bGMM and GNAs.

**Strengths:**

Data augmentation plays an important role in deep learning. The paper provides a theoretical analysis of using generative models for data augmentation. Understanding generative data augmentation can potentially benefit machine learning tasks in low-data conditions.

**Weaknesses:**

- In Section 4.2, the authors use standard augmentation to approximate CIFAR-10 with larger $m_S$. The comparison of GDA on CIFAR-10 and augmented CIFAR-10 is unfair as the standard augmentations induce effective inductive bias in the dataset, e.g., flipping. A more proper way to approximate CIFAR-10 with different $m_S$ is to sample multiple subsets of CIFAR-10 data with different sizes.

- Although the paper proposes a general stability bound on GDA, it is unclear how the stability bound can be utilized to advance existing baselines. The finding that GDA can improve test generalizations on small datasets, where awful overfitting occurs, is somewhat expected. It is still unclear how to set the hyperparameters, like the number of augmented samples given any new dataset.


**Questions:**

- How to use the stability bound to advance existing generative data augmentation methods in practice?

**Limitations:**

I find no negative societal impact in this work.

---

> ### Author Rebuttal · Authors · 2023-08-09
>
> # Response to Reviewer hFUC
>
> We thank Reviewer hFUC for the valuable comments.
>
> ## Weakness 1: Standard augmentation
>
> Thanks for the helpful advice. **There is no comparison between the CIFAR-10 and the augmented CIFAR-10, and the standard augmentation is used to approximately verify our theory with large $m_S$**. Our experimental results support the theoretical results, which is agreed by Reviewer SG4V and jB6r.
>
> * "Unfair comparison between CIFAR-10 and augmented CIFAR-10". **We note there is no comparison between the CIFAR-10 and the augmented CIFAR-10**. They are two different cases where we want to verify Theorem 3.3, respectively.  First, when standard augmentation is not used, the $m_S$ is small. In this case, we validate that GANs can improve the test performance (see Remark on line 280). Second, when standard augmentation is approximately used, the $m_S$ is approximately large. In this case, we validate that GANs can not boost the test performance obviously and even damage it (see Remark on line 276).
> * "Approximating CIFAR-10 with different $m_S$". When $m_S \leq 50,000$ (the size of the CIFAR-10 dataset), we agree that sampling the subset of the CIFAR-10 dataset is the correct method. However, in that case, we can only observe worse overfitting, **which fails to verify our theoretical results that GANs can not improve the test performance when $m_S$ is large**. Therefore, we must investigate the setting where $m_S > 50,000$. However, when $m_S > 50,000$, **simply oversampling will cause the repeat of many data points**, so we chose the common augmentation for CIFAR-10 [51] to approximate the large $m_S$.
>
> We will discuss these in detail in the final version.
>
> ## Weakness 2: Expected results on small dataset & Optimal augmentation size
>
> Thanks for the suggestion. We discuss the "expected results" and the setting of hyperparameters respectively.
>
> ### "Expected results"
>
> **Precisely analyzing the expected phenomena, in theory, contributes to the community**. These expected results empirically validate the proposed theoretical framework. Please see the details in our response to the common concern 2.
>
> ### Setting of hyperparameters
>
> Our work serves as a theoretical foundation to be extended in the future, which is agreed by Reviewer SG4V and jB6r. It gives insight to the optimal augmentation size $m_G^*$. Please see details in our response to the common concern 1.
>
>
> ## Q1: How to use the stability bound to advance existing generative data augmentation methods in practice
>
> Thanks for the suggestion. This paper is mainly a theoretical work and a first step towards understanding the GDA, so it is hard to use it to guide the practice detailedly. However, our results can still give some insights to the practice. Please see details in our response to the common concern 3.
>
> We will discuss the impact on the practice detailedly in the final version.
>
> **We think we have fully addressed the questions from the reviewer. If the reviewer has any further questions, please feel free to contact us for further discussion**.

---

> > ### Comment · Reviewer_hFUC · 2023-08-14
> >
> > Thank you for the response. The authors addressed my primary concerns and provided some insights into how the stability bound can be used in practice. Therefore, I am increasing my score from 4 to 5.

---

> > > ### Author Response · Authors · 2023-08-14
> > > **Official Comment by Authors**
> > >
> > > Thanks very much for your valuable comments and the update on the rating.

---

### Official Review · Reviewer_jB6r · 2023-07-03

**Soundness:** 3 good
**Presentation:** 3 good
**Contribution:** 4 excellent
**Rating:** 7
**Confidence:** 4

**Summary:**

This paper studies generative data augmentation (GDA), in which the samples from trained generative models are added to the training dataset for training discriminative models. There have been several empirical research reports on GDA, and it is known that GDA is unlikely to be effective when real learning data is abundant. However, there has been no theoretical analysis provided so far to explain this phenomenon. This study assumes a realistic setting in which the distribution imitated by the generative model and the real dataset distribution are different, and analyzes the generalization error bounds of GDA in the non-iid setting in three cases: (i) general case based on the existing algorithmic stability framework, (ii) a binary Gaussian mixture model (bGMM) and linear classifier, and (iii) a deep generative adversarial nets and deep neural classifier. The main findings from these theorems are (a) the divergence between the distribution imitated by the generated model and the true distribution is important for the generalization error, (b) increasing the number of generated samples does not lead to a faster learning rate, and (c) GDA does not achieve faster learning rate in situations affected by the curse of dimensionality. The paper provides simple experiments on bGMM and CIFAR-10 to test the theory, showing the similarity between the upper bound of the generalization error given by the theorem and the measured trends of the error, and the effectiveness of GDA in situations where the curse of dimensionality occurs. While this generalization error analysis is not perfect, as the paper states in the Limitation, this paper will have a significant impact on the research field of GDA, where no theoretical discussion existed.

**Strengths:**

+ The paper is well organized and clearly states its arguments. Also, the paper is very readable, with careful notation and explanation of existing frameworks to explain the theory.
+ The paper provides generalized error-bound analysis in three settings ranging from general to realistic settings, establishing a first step toward learning guarantees in GDA.
+ The paper confirms the implications of the theorem through several experiments.

**Weaknesses:**

- **W1** Some parts of the explanation are difficult to interpret. In Theorem 3.2 and 3.3, the paper explains "constant-level improvement" by GDA, but it is not clear in which equation "constant" appears, making the argument difficult to understand. Further, in Eq. (2), most of the theorems depend on the explanation in the Appendix, and the paper is not self-contained in this respect.
- **W2** The paper does not mention several important previous studies. For example, Shmelkov et al. [a] were the first to report that GDA with GANs degrades accuracy even in small training dataset settings. Subsequently, Tran et al. [b] and Yamaguchi et al. proposed methods to improve GDA by the principles of Bayesian neural networks and multi-task learning. Although these works use somewhat older generative models and there may be facts that partially contradict the claims of the paper, they are considered important milestones in explaining the motivation for your work. I recommend that the paper cites them appropriately.

**Reference**
- [a] Shmelkov, Konstantin, Cordelia Schmid, and Karteek Alahari. "How good is my GAN?." Proceedings of the European conference on computer vision (ECCV). 2018.
- [b] Tran, Toan, et al. "A bayesian data augmentation approach for learning deep models." Advances in neural information processing systems 30 (2017).
- [c] Yamaguchi, Shin'ya, Sekitoshi Kanai, and Takeharu Eda. "Effective data augmentation with multi-domain learning gans." Proceedings of the AAAI Conference on Artificial Intelligence. Vol. 34. No. 04. 2020.

**Questions:**

- **Q1** Can the theorems provided by the paper discuss GDA in the situation of transfer learning of generative models? We often apply transfer learning such as fine-tuning to compensate for the distribution approximation performance of the generative model in GDA. I understood that minimizing distribution's divergence by transfer learning is not matter because Theorem 3.1 makes no assumptions about training methods, is this correct?
- **Q2** Why is GAN set as the generative model in Theorem 3.3? Is there any advantage in proving this theorem?
- **Q3** This is mostly a comment, but I think that the experiments in Section 4.2 should be included in the main paper since the evaluation of GDA with diffusion models as well as GANs is a high-impact result.

**Limitations:**

This paper adequately discusses the limitation of the tightness of the theoretical guarantee.

---

> ### Author Rebuttal · Authors · 2023-08-09
>
> # Response to Reviewer jB6r
>
> We thank Reviewer jB6r for the acknowledgment to our contributions and insightful and constructive comments.
>
> ## Weakness 1: Constant-level improvement
>
> Thanks for the nice advice. In general, given a fixed $m_G$, it is challenging to obtain an explicit form of "constant-level improvement" due to the complexity of the generalization bound. However, **we can clarify the "constant-level improvement" more clearly by diving into the explicit bounds in Appendix** (e.g., Eq. (19)). We denote the generalization error bound by ${Error}(m_G)$, where $m_G$ is the augmentation size. We compare the cases where $m_G = 0$ (without GDA) and $m_G \to +\infty$, respectively. Then the following holds.
>
> * bGMM: when $d > m_S$, we have $Error(+\infty) \leq \frac{1}{\log(m_S)} Error(0)$.
>
> * GANs: when $\sqrt{d} > m_S$, we have $Error(+\infty) \leq \frac{1}{AL^2} Error(0)$, where $A = \prod_{l} \Vert W_l \Vert$.
>
> These results can be proved by plugging $m_G = 0$ and $m_G \to +\infty$ into the bounds in the proof of Theorem 3.2&3.3. **We will re-organize these theorems and add these results and discussion in the final version**. In addition, we visualized the explicit bound of bGMM setting in Figure 1 d&e&f, which empirically validates the "constant-level improvement".
>
> ## Weakness 2: Need to cite some previous works
>
> Thanks for the helpful suggestion. We will cite the mentioned papers in the final version.
>
> ## Q1: Transfer learning of generative models
>
> Thanks for the insightful comment. Yes, if we are interested in analyzing the case where some transfer learning techniques are used to improve the distribution approximation performance of the generative model (decreasing $d_{TV}(D, D_G(S))$), our results (Theorem 3.1) can be a theoretical foundation. By decreasing $d_{TV}(D, D_G(S))$, our results show that GDA can perform better when a suitable augmentation number is chosen. We will add more discussion to the conclusion in the final version.
>
> ## Q2: Why does theorem 3.3 choose GAN
>
> Thanks for the comment. In this paper, we use the existing results (Lemma B.12) of GANs [46] to bound the $d_{TV}(D, D_G(S))$ in Theorem 3.3. With the emergence of more advanced results of generative models, our results can be improved and extended further.
>
> ## Q3: Some important results in Appendix
>
> Thanks. We will move Table 2 to the main paper in the final version following your suggestion.

---

> > ### Comment · Reviewer_jB6r · 2023-08-12
> >
> > Thank you for providing the rebuttal. The authors adequately addressed my concerns and promise to revise the paper according to the response. So, my confidence level regarding my review of this paper has increased. Thank you.

---

> > > ### Author Response · Authors · 2023-08-12
> > >
> > > Thanks again for your valuable comments and acknowledgment of our work.

---

### Official Review · Reviewer_Dxq5 · 2023-07-06

**Soundness:** 3 good
**Presentation:** 2 fair
**Contribution:** 3 good
**Rating:** 5
**Confidence:** 3

**Summary:**

Generative data augmentation (GDA) aims to improve model performance by generating artificial labeled samples to enlarge the limited training dataset, but is also highly influenced by the size of training dataset, choices of augmentation methods and the number of augmented data. The paper seeks to develop a theoretical understanding to GDA by reproducing classical results built upon i.i.d. assumptions to the generatively augmented dataset, and establishes a general relationship between the stability bound and the learned distribution divergence besides the augmentation size. To further investigate and verify the proposed upper bound of generalization error, the paper also particularizes specific cases of bGMM and GAN, and provides insights to understand the theoretical results. Finally, empirical experiments on synthetic dataset and CIFAR10 are conducted to study the effect of different factors and validate discovered theoretical findings.

**Strengths:**

1） Mathematical notations and theoretical assumptions are clearly exhibited and explained in the paper. Details required to understand the problem background and existing results are also provided.
2） The theoretical results are novel and the related analysis seems to be solid and intuitive. The proposed theorems also build connection to previous results under i.i.d. assumption and empirical findings.
3） The idea of extending classical generalization error theories to the generative data augmentation problem is novel and well-motivated.


**Weaknesses:**

1） Despite theoretical results and corresponding conclusions of this paper are easy to understand and conformable to our common intuition, too many separated compositions of remarks make the paper annoyingly disorganized, forcing the readers to jump around while reading. The paper has to be re-organized to make readers easier to access the content.

2） Some figures (e.g. Figure 1a) are hard to recognize, and some are unnecessarily separated into parallel parts (e.g. Figure 1b and Figure 1c).

3） My greatest concern about this work is that when talking about data augmentation, we are most interested in the difference of model performance trained with and without data augmentation. However, the paper only considers the case where the optimal augmentation number $m_G^*$ is adopted.

4） According to Theorem 3.1, the generalization bound for GDA is composed of the distribution divergence and the generalization error with respect to the mixed distribution, where the former is determined by the model itself and the latter is controlled by the number of generative data. For a given model with fixed generative ability, what we can only do is to find a near optimal number of generative samples. However, the choice of this number is not inspiring in this paper in my opinion.

5） Experiments on real-world dataset seem incomplete and not convincing enough. First, only a single dataset (CIFAR-10) is adopted which fails to evade possible influence by the nature of dataset. Second, the experimental results show limited information and connection to the proposed theories, since the generalization error is inestimable in this case. Moreover, it is unfair to take generative models with totally different architectures for comparison, but fair comparison can be made between the same generative model with different training degrees.

6） The practical contribution of the paper may be limited since many conclusions are intuitive and show little help to practical application.


**Questions:**

1) In the paper the mixed distribution with augmentation is simply defined as a weighted combination of the training set distribution and the generative model distribution, is it a proper definition particularly when these two distributions are mutually dependent?
2) Could the authors clarify the purpose of choosing DCGAN, StyleGAN and EDM for experiments since there are plenty of alternative choices such as classical variational autoencoder (VAE)? This part is unclear for me from the paper.
3) The theoretical results and conclusions given by the paper are solid. But could the authors illuminate what can be inspired from these conclusions especially for utilizing generative data augmentation in practical applications?


**Limitations:**

see above

---

> ### Author Rebuttal · Authors · 2023-08-09
>
> # Response to Reviewer Dxq5
> We thank Reviewer Dxq5 for the valuable comments.
> ## Weakness 1: Organization
> Thanks for the advice. We will re-organize the separated remarks in a coherent and integrated manner.
> ## Weakness 2: Figures
> We will make the figures more recognizable by removing some unimportant lines. Besides, we will integrate Fig. 1b and Fig. 1c (similarly, Fig. 1e and Fig. 1f) into a double y-axis graph.
> ## Weakness 3:  Difference between with and without GDA & Only considers the case with $m_G^*$
> Thanks for the suggestion. We discuss two concerns respectively.
> ### Difference between with and without GDA
> This is one of the main questions we want to answer in this paper. It can be divided into two cases, one with large $m_S$ and the other with small $m_S$, detailed as follows.
> * Large $m_S$. In this case, $m_S$ dominates the generalization bound. In Corollary 3.1, we conclude that if $d_{TV}(D,D_G(S))=o (\max(\log(m)\beta_m,1/\sqrt{m}))$, then using the GDA enjoys a faster learning rate than not using the GDA. Besides, in both the bGMM and GANs setting (Theorem 3.2&3.3), we prove the precondition fails to hold, so the GDA is ineffective when $m_S$ is enough.
> * Small $m_S$. In this case, some terms (e.g. dimension) dominate the generalization bound, and awful overfitting happens. In the bGMM and GANs settings (Theorem 3.2&3.3), we prove that GDA can bring a constant-level improvement to the generalization performance.
> ### Only considers the case with $m_G^*$
> **In fact, we do not only consider the case with $m_G^\*$**. The choice of $m_G$ can be divided into 4 cases:
> * $m_G=0$. It means that the GDA is not used, which is the baseline we want to compare.
> * $m_G=\Theta(m_S)$. It is the common case when people use the GDA.
> * $m_G=m_{G,order}^*$. It is the efficient augmentation size that achieves the fastest learning rate w.r.t. $m_S$.
> * $m_G=m_{G}^*$. It is the optimal augmentation size that minimizes the generalization bound in Theorem 3.1.
>
> **All the main theorems** in the paper include (Theorem 3.1) or highlight (Theorem 3.2&3.3) the first three settings. Besides, **all experiments** also mainly focus on the cases where $m_G=0$ or $m_G=\Theta(m_S)$ ($m_G\leq50m_S$ for the bGMM and $m_G\leq20m_S$ for GANs).
>
> We will discuss these more detailedly in the final version.
> ## Weakness 4: Optimal augmentation size
> Thanks for the suggestion. Our work serves as a theoretical foundation and gives insights to the optimal augmentation size. Please see details in our response to the common concern 1.
> ## Weakness 5: Experimental design
> Thanks for the advice. **We adopt the CIFAR-10 dataset to empirically verify our theory, where the generalization error can be estimated and no comparison exists between different generative models**. Our experimental results support the theoretical results, which is agreed by Reviewer SG4V and jB6r.
> * "Only a single CIFAR-10 dataset is adopted". CIFAR-10 is a widely used dataset and **we adopt it to empirically validate Theorem 3.3**. Combining the simulations in the bGMM setting, our theory is verified sufficiently.
> * "The generalization error is inestimable in the CIFAR-10 dataset". By definition, given a trained neural classifier, the generalization error of Theorem 3.3 can be estimated by the absolute gap between the mean cross-entropy loss on the training set (with generated data) and the mean cross-entropy loss on the test set. **We add the results of GANs with this estimator in the latest uploaded PDF (Table A)**. On the one hand, GANs decrease the generalization error when $m_S$ is small (without standard augmentation). On the other hand, GANs fail to boost the performance obviously and even hurts the error when $m_S$ is approximately large (with standard augmentation). **The results support Theorem 3.3 again**.
> * "Unfair to take generative models with totally different architectures for comparison". **The experiments are conducted to verify our Theorem 3.3, rather than comparing different generative models**. How these generative models verify our theory can be found in our response to Q2. **To reduce the confusion here, we will split Table 2 into three tables according to the generative models**.
> We will clarify these more clearly in the final version.
> ## Weakness 6:  Intuitive results show little help to practice
> Thanks for the advice. Analyzing the intuitive phenomena contributes to the community. Please see details in our response to the common concern 2.
> ## Q1: Definition of the mixed distribution
> Yes, it is a proper definition no matter whether the $D_G(S)$ is dependent on the $D$. In fact, the convex combination of arbitrary distributions is still a distribution. Welcome to ask further questions if you still have confusion.
> ## Q2: The purpose of choosing DCGAN, StyleGAN, and EDM
> Thanks for the suggestion. **GANs are chosen to empirically validate Theorem 3.3 and the EDM is chosen to explore the ability of the diffusion model.** First, we choose a "bad" GAN (DCGAN) to empirically verify that GANs can improve the test performance when awful overfitting happens (without standard augmentation). Second, we choose a "good" GAN (StyleGANv2) to verify that GANs can not improve the test performance obviously when the $m_S$ is approximately large (with standard augmentation). Third, because diffusion models achieve good success in recent years, we conduct experiments on the EDM and suggest that diffusion models have a better $d_{TV}(D, D_G(S))$ than GANs. We will discuss these more detailedly in the final version.
> ## Q3: Impact on the practice
> This paper is a first step towards understanding the GDA, so it is difficult to guide the practice detailedly. However, our results can still give some insights to the practice. Please see details in our response to the common concern 3.
>
> **We think we have fully addressed the concerns of the reviewer. If the reviewer has any further questions, please feel free to contact us for a further discussion**.

---

> > ### Comment · Reviewer_Dxq5 · 2023-08-18
> >
> > Thanks for providing the detailed rebuttal. The authors' responses address most of my concerns, hence I raised my rating to borderline acceptance.

---

> > > ### Author Response · Authors · 2023-08-18
> > >
> > > Thanks very much for your valuable comments and the update on the rating!

---

### Official Review · Reviewer_SG4V · 2023-07-11

**Soundness:** 3 good
**Presentation:** 3 good
**Contribution:** 3 good
**Rating:** 7
**Confidence:** 1

**Summary:**

In this work the authors present new theoretical results for generative data augmentation. In particular, the authors introduce a new result that gives a bound on the generalization error of a model trained with data augmentation provided by a generative model. The authors use this bound to illustrate when GDA may or may not help in generalization performance. The authors then provide specific examples of applications of this bound to common models for generative augmentation; a binary Gaussian mixture and a deep generative model (GAN). For each the authors derive a more specialized bound and perform empirical experiments to validation the theory.

**Strengths:**

- This work provides useful theoretical insight into the strengths and limitations of generative data augmentation. The authors present novel results that, to my knowledge, are the first results bounding the generalization error for models trained with the aid of generative data augmentation.
- I have not checked proofs in the appendix, but the theory as presented in the main text seems sound.
- The applications to both the binary Gaussian mixture and GAN setting are helpful for illustrating and expanding upon the main theoretical results.
- The experimental results do support the theoretical results presented and the results on DGM showing the re-affirming the promise of diffusion models for this purpose are interesting.
- The results presented could be a useful foundation for future work.

**Weaknesses:**

- The deep generative model results presented are narrow in scope, applying to GANs trained in a class-separated manner and to fully-supervised learning, rather than potentially semi-supervised learning or transfer learning via generative models.
- As mentioned by the authors, they do not investigate the tightness of the proposed bounds.
- I'm not sure this type of data augmentation approach is widely used enough for this analysis to be immediately impactful, though it does appear to be gaining traction.
- The clarity of the writing could be improved in places.

**Questions:**

- I was very confused by the usage of the term "learning rate" in the context here. I think I now understand it to refer to rate at which the generalization bound shrinks as a function of the amount of data, but given that the paper also discusses SGD, where the term has a different meaning it's hard to follow. Or is there some connection that I'm not appreciating?
- I'm similarly a bit confused by what is meant by "augmentation consumption" on line 166. Can you elaborate on this?
- I found the overloading of $d$ as both the data dimension and as a way to denote a divergence confusing, particularly for theorem 3.2.
- Can you offer further any insight into when and why GDA does help in practice?

**Limitations:**

The authors discuss limitations.

---

> ### Author Rebuttal · Authors · 2023-08-09
>
> # Response to Reviewer SG4V
>
> We thank Reviewer SG4V for the positive score and valuable comments.
>
> ## Weakness 1: Scope of  Theorem 3.3
>
> Thanks for the helpful suggestion. Though we choose specified GANs and supervised learning in Theorem 3.3, **our general framework (Theorem 3.1) is a foundation and can be extended to other generative models and training methods**.
>
> * "GAN". The proposed framework (Theorem 3.1) can also be used to analyze other generative models, as long as we can estimate the  $d_{TV}(D, D_G(S))$, and $\mathscr{T}(m_S, m_G)$ in the concrete case.
> * "Class-separated manner". Deriving the generalization bound ($d_{TV}(D, D_G(S))$) for the conditional generative models is still challenging in the literature. As a preliminary work to understand the GDA, we assume the "class-separated manner" to simplify the analysis. With the emergence of better analysis for the conditional generative models, our results can also be further improved.
> * "Fully-supervised learning".  As a first step towards understanding the GDA, it is reasonable to investigate the basic supervised classification setting. Besides, the non-i.i.d. result of supervised learning is a starting point, which can inspire the derivation of other training methods (e.g. semi-supervised learning).
>
> We will clarify these more clearly in the final version.
>
> ## Weakness 2: Tightness of our bounds
>
> Thanks for the suggestion. In general, finding the lower bounds for the stability generalization bounds is still an open topic (e.g., see [23]).  In this paper, **our results are tighter than using the existing non-i.i.d. bounds directly** (see Appendix C). The lower bounds can be left to future works. We will discuss this more detailedly in the Limitations section.
>
> ## Weakness 3: Conditional data augmentation
>
> Thanks for the nice suggestion. **Class-conditional generative data augmentation has been widely used**, and can empirically improve performance in lots of settings, including supervised learning [13, 14], semi-supervised learning [15, 16, 17], few-shot learning [18], zero-shot learning [19], adversarial robust learning [20, 21], etc (see line 25-26).
>
> ## Weakness 4: Clarity of the writing
>
> Thanks for the helpful suggestion. We will try our best to improve the clarity of the writing in the final version.
>
> ## Q1: "Learning rate"
>
> Thanks for the helpful advice. Yes, in this paper **except line 258**, it refers to the rate at which the generalization bound shrinks as a function of the amount of data ($m_S$ in this paper). On line 258, it means the step size of the SGD. We will avoid this ambiguity in the final version by **replacing the "learning rate" on line 258 with "step size"**.
>
> ## Q2: "Augmentation consumption"
>
> Thanks for the suggestion. The consumption can be divided into three parts. First, **sampling consumption**. Sampling more data will need more computation. Second, **store consumption**. More sampled data will need more memory to store. Third, **training consumption**. Given more generated data, if we fix the number of training epochs, the training of the downstream tasks will take more time. We will clarify this definition in the final version.
>
> ## Q3: Overloading of $d$
>
> Thanks for the suggestion. We will avoid the ambiguity in the final version by **using $\mathcal{D}$ to denote the divergence**.
>
> ## Q4:  When and why GDA does help in practice?
>
> Thanks for the suggestion. It can be divided into two cases, one with large $m_S$ and the other with small $m_S$ (compared with other terms in the bound), detailed as follows.
>
> * Large $m_S$. In the Corollary 3.1, we conclude that if $d_{TV}(D, D_G(S)) = o \left(\max (\log(m)\beta_m, 1/ \sqrt{m}) \right)$, then using the GDA enjoys a lower generalization error than not using the GDA, which is because the generative model learns faster than the classifier. In both the bGMM and GANs settings, we prove that $d_{TV}(D, D_G(S)) = \Omega \left(\max (\log(m)\beta_m, 1/ \sqrt{m}) \right)$, so the GDA (with arbitrary $m_G$) is ineffective when $m_S$ is enough (Theorem 3.2&3.3).
> * Small $m_S$. In this case, other terms (e.g. dimension) dominate the generalization bound, and awful overfitting happens. In the bGMM and GANs settings, we prove that GDA can bring a constant-level improvement to the generalization performance (Theorem 3.2&3.3). This is because more data are significant to relieving the overfitting.
>
> We will discuss this more detailedly in the final version.

---

> > ### Comment · Reviewer_SG4V · 2023-08-16
> > **Reply to authors**
> >
> > Thank you for your thoughtful responses! Reading through this and the other responses has convinced me to bump up my score. I agree that this seems to be an interesting step towards analyzing the performance of generative data augmentation. My confidence is still low as I am pretty unfamiliar with related work on generalization bounds.

---

> > > ### Author Response · Authors · 2023-08-16
> > > **Official Comment by Authors**
> > >
> > > Thanks very much for your valuable comments and the update on the rating!

---

### Author Rebuttal · Authors · 2023-08-09

# Summary of the revision

We sincerely thank the reviewers for their valuable comments, which help to further improve the quality of our work. We have thoroughly addressed the detailed comments. and summarize the revision in the next version as follows:

## New results

* **We add the results of GANs with the estimated generalization error in the latest uploaded PDF (Table A)**.
* We add more theoretical results to clarify the constant-level improvement in Theorem 3.2&3.3.

## Writing

* We avoid the overloading of $d$ by using $\mathcal{D}$ to denote the divergence.
* We clarify the meaning of "augmentation consumption" on line 166.
* We replace the "learning rate" on line 258 with "step size" to avoid ambiguity.
* We make the figures more recognizable by removing some unimportant lines. Besides, we integrate Fig. 1b and Fig. 1c (similarly, Fig. 1e and Fig. 1f) into a double y-axis graph.
* We move Table 2 to the main paper.

## Discussion

* We will cite the papers mentioned by Reviewer jB6r.
* We add more discussion about the scope of the theoretical results.
* We add more discussion about the design of our experiments on the CIFAR-10 dataset and its relation with Theorem 3.3.
* We add more discussion about the impact of work on the practice.
* We discuss the tightness of our bounds more detailedly in the Limitations section.

# Common concerns from reviewers

We thank all reviewers for their valuable and constructive comments. We address the common concerns here and post a point-to-point response to each reviewer as well. We believe the quality of the paper has been improved following the reviewers' suggestions.

## Common concern 1:  Choice of  the optimal number of generated samples (from reviewer Dxq5 and hFUC)

Our work serves as a theoretical foundation to be extended in the future, which is agreed by Reviewer SG4V and jB6r. The optimal augmentation size $m_G^*$ can be decided when $d_{TV}(D, D_G(S))$, $\beta_m$ and $\mathscr{T}(m_S, m_G)$ are estimated in the concrete situation. With the emergence of more advanced theory for the generative models, $m_G^*$ will be estimated better in the future. The reviewers can refer to the response to Common concern 3 for a more detailed discussion of our impact on the practice.

## Common concern 2: Intuitive/expected results show little help to practical application (from reviewer Dxq5 and hFUC)

**Precisely analyzing the expected phenomena, in theory, contributes to the community**. These expected results are predicted well by our theoretical results, so they empirically validate the proposed framework. We also clarify things more clearly from a theoretical perspective. For example, we prove that the size of a "small dataset" is relative to the data dimension and that the improvement brought by the GDA is constant-level rather than order-level. The reviewers can refer to the response to Common concern 3 for a more detailed discussion of our impact on the practice.

## Common concern 3:  Impact on the practice (from reviewer Dxq5 and hFUC)

This paper is mainly a theoretical work and a first step towards understanding the GDA, so it is still difficult to use it to guide the practice detailedly. However, our results can still give some insights to the practice, detailed as follows.

* Theorem 3.1 implies that improving the distribution approximation performance of the generative models is important for the GDA, which decreases the generalization error by optimizing the $d_{TV}(D, D_G(S))$. This motivates people to design better generative models.
* Theorem 3.1 shows that stabilizing the training of the generative models can bring benefits to the GDA, which optimizing the term $\mathscr{T}(m_S, m_G)$ and thus reduce the generalization error. This motivates us to improve the stability of the training of generative models (e.g. GAN). Besides, some transfer learning techniques can be used to optimize the $d_{TV}(D, D_G(S))$, which is mentioned by Reviewer jB6r.
* Theorem 3.1 implies that if we can estimate $d_{TV}(D, D_G(S))$, $\beta_m$ and $\mathscr{T}(m_S, m_G)$ for the concrete case, then the optimal augmentation size can be decided. With the emergence of more advanced theory for the generative models, $m_G^*$ will be estimated better in the future.
* When $m_S$ dominates the generalization bound, Theorem 3.1 give us a sufficient condition on when GDA works. If $d_{TV}(D, D_G(S)) = o \left(\max (\log(m)\beta_m, 1/ \sqrt{m}) \right)$, then using the GDA enjoys a lower generalization error than not using the GDA. Specially, in both the bGMM and GANs setting (Theorem 3.2&3.3), we prove that $d_{TV}(D, D_G(S)) = \Omega \left(\max (\log(m)\beta_m, 1/ \sqrt{m}) \right)$, so the GDA (with arbitrary $m_G$) is ineffective when $m_S$ is enough or standard augmentation is used.
* When other terms (e.g. dimension) dominate the generalization bound, awful overfitting happens. In the bGMM and GANs settings (Theorem 3.2&3.3), we prove that GDA can bring an improvement to the generalization performance. Specially, we prove the improvement is constant-level rather than order-level.

With the emergence of more advanced theories, our results can be improved and give more guidance to the practice. We will discuss the impact on the practice more detailedly in the final version.

---

### Decision · Program_Chairs · 2023-09-21

**Decision:**

Accept (poster)

**Comment:**

The paper presents a theoretical analysis of generative data augmentation (GDA), a technique that uses generative models to create synthetic data that is added to the training set to improve the generalization performance of machine learning models. The authors provide theoretical guarantees on the performance of GDA under different assumptions on the generative model and the data distribution. They also conducted experiments on the CIFAR-10 dataset to demonstrate the effectiveness of GDA.

The paper is well-written and the presentation is clear. The authors do a good job of explaining the technical details in a way that is accessible to a general audience. The experiments are well-designed and the results are convincing to the reviewers who all raised their grades after the rebutal.

Overall, the paper is a valuable contribution to the literature on generative data augmentation. It provides a theoretical understanding of the benefits of GDA and insights on how to choose the optimal augmentation size.

Here are some specific points that I liked about the paper:

The authors provide a clear and concise overview of the GDA technique.
The theoretical analysis is rigorous and provides insights on the benefits of GDA.
The experiments are well-designed and the results are convincing.
The paper is well-written and the presentation is clear.
I would like to see the authors address the following points in the final version of the paper:

The authors could provide more discussion on the impact of GDA on the practice. For example, they could discuss how GDA can be used to improve the performance of machine learning models on different tasks.
The authors could also provide more discussion on the limitations of GDA. For example, they could discuss how the performance of GDA depends on the quality of the generative model.
Overall, I think this is a well-written and well-researched paper that makes a significant contribution to the literature on generative data augmentation. I recommend it for publication.